# Characterizing basin-scale precipitation gradients in the Third Pole region using a high-resolution atmospheric simulation-based dataset

Yaozhi Jiang[1], Kun Yang[1, 2, *], Hua Yang[2], Hui Lu[1], Yingying Chen[2], Xu Zhou[2], Jing Sun[1], Yuan Yang[3], Yan Wang[4]

5    [1] Department of Earth System Science, Ministry of Education Key Laboratory for Earth System Modeling, Institute for Global Change Studies, Tsinghua University, Beijing, China.

[2] National Tibetan Plateau Data Center, State Key Laboratory of Tibetan Plateau Earth System, Environment and Resources, Institute of Tibetan Plateau Research, Chinese Academy of Sciences, Beijing, China.

10    [3] Institute of Science and Technology, China Three Gorges Corporation, Beijing, China

[4] Key Laboratory of Land Surface Pattern and Simulation, Institute of Geographic Science and Natural Resources Research, Chinese Academy of Sciences, Beijing 100101, China

**Corresponding author:** Kun Yang (yangk@tsinghua.edu.cn)

**Abstract:** Altitudinal precipitation gradient plays an important role in the interpolation of precipitation in the Third Pole (TP) region, where the topography is very complex but in-situ data are very sparse. This study proves that the altitude dependence of precipitation in the TP can be reasonably reproduced by a high-resolution atmospheric simulation-based dataset called ERA5_CNN. The precipitation gradients, including both absolute (APG) and relative gradients (RPG), for 388 sub-basins of the TP above 2500 m a.s.l. are calculated based on the ERA5_CNN. Results show that most sub-basins have positive precipitation gradients, and negative gradients are mainly found along the Himalayas, the Hengduan Mountains and the Western Kunlun. The annual APG and RPG averaged across all sub-basins of the TP are 0.05 mm.day$^{-1}$.100 m$^{-1}$ and 4.25 %.100 m$^{-1}$, respectively. The values of APG are large in wet seasons but small in dry seasons, while the RPG shows opposite variations. Further analyses demonstrate that the RPGs have negative correlations with relative humidity but positive correlations with wind speed, likely because dry air tends to reach saturation at high altitudes, while stronger wind can bring more humid air to high altitudes. In addition, we find that precipitation gradients tend to be positive at small spatial scales compared to those at large scales, mainly because local topography plays a vital role in determining precipitation distribution at small scales. These findings on the spatiotemporal variations of precipitation gradients provide useful information for interpolating precipitation in the TP.

**Keywords:** precipitation gradient; the Third Pole; high-resolution atmospheric simulation; spatiotemporal variation

## 1. Introduction

Gridded precipitation is a key input for many hydrological and ecological models when applied to regional studies. Typically, the spatial distribution of precipitation in a region can be obtained by interpolating the in-situ data. In regions with flat terrain and dense rain gauge networks, different interpolation methods (e.g. Thiessen polygons, inverse distance weighting, Kriging) can result in similar distributions of precipitation. In mountainous regions, precipitation has great spatial heterogeneity and sparse rain gauges with limited spatial representativeness make the interpolation of precipitation challenging in these regions. Relations between precipitation and other environmental factors (e.g. topography and vegetation) play an important role in the interpolation of precipitation, especially in mountainous regions. Among the many environmental factors, altitude has a significant impact on the distribution of precipitation. Several widely-used interpolation models have taken altitude as a covariant, such as PRISM (Daly et al., 1997) and ANUSPLIN (Hutchinson, 1991). Therefore, quantifying the precipitation gradient is greatly important in mountainous regions.

As the main source of many large rivers in Asia, the Third Pole (TP) is a typical mountainous region in the world, characterized by complex terrain and high altitude. Rain gauges in the TP are sparse and usually located in lowland areas, where the weather conditions are much different from those in high altitudes (Chen et al., 2012; Daly et al., 2002). Therefore, interpolating in-situ data to data-sparse high altitudes is essential for hydrometeorological studies in this region, as reported in many previous studies that taking the precipitation gradient into account in hydrological modeling results in better simulations (Immerzeel et al., 2014; Li Wang et al., 2018; Zhang et al., 2015). Currently, studies on the altitude dependence of precipitation are mostly in the eastern TP (Cuo and Zhang, 2017; Guo et al., 2016) and some sub-regions, such as the Himalaya (Ouyang et al., 2020; Salerno et al., 2015; Yang et al., 2018), the Qilian Mountains (Chen et al., 2018; Lei Wang et al., 2018), the Yarlung Tsangpo River Basin (Sun and Su., 2021) and the Hengduan Mountains (Yu et al., 2018). Moreover, the altitudinal precipitation gradients obtained in these studies are usually based on rain gauge data, which may misrepresent precipitation gradient due to the poor representativeness of rain gauges. For most parts of the TP, particularly the central and western TP, the precipitation gradient remains unknown. Besides, the precipitation gradient may vary with different seasons and years due to the changes in weather and meteorological conditions, and the temporal variability of the precipitation gradient in the TP has not been investigated yet.

In previous studies, satellite precipitation products have also been used to calculate the precipitation gradient in the TP (Liu et al., 2011). However, satellite products tend to contain large

uncertainties and are less accurate in complex-terrain regions (Derin and Yilmaz, 2014; Henn et al., 2018; Shen et al., 2014; Xu et al., 2017). In the western TP, where solid precipitation is dominated, the satellite products cannot reproduce the actual spatial variability of precipitation (Li et al., 2020). Therefore, obtaining the precipitation gradient based on satellite products seems to be undesirable in these regions.

Recently, high-resolution atmospheric simulations have made great progress in the TP and its surroundings and many atmospheric simulation-based precipitation datasets have been arising (Maussion et al., 2014; Pan et al., 2012; Y. Wang et al., 2020; Zhou et al., 2021). The atmospheric simulations are constrained by a set of physical processes and thus can well represent the influence of topography on precipitation distribution when integrated with high resolution (Lundquist et al., 2019; Y. Wang et al., 2018). Previous studies have demonstrated the potential of atmospheric simulations (especially convective-permitting simulations) in capturing spatial variability of precipitation in the TP, e.g. Zhou et al. (2021) found that the dynamically downscaled precipitation of ~3 km horizontal resolution has high correlations with observations in the TP; Gao et al. (2020) found that a convective-permitting simulation could better reproduce the precipitation distribution and further result in better snow cover simulation than satellite-based products in the southeastern TP. Similar results were also reported in the Himalayas (Collier and Immerzeel, 2015; Ouyang et al., 2021) and western TP (Pritchard et al., 2019). These studies indicate that high-resolution atmospheric simulations can be alternative sources for obtaining the precipitation gradient in the TP, particularly in regions like the western TP with almost no rain gauges located.

Therefore, the main objective of this study is to obtain the altitudinal precipitation gradient for different sub-basins of the TP based on a high-resolution atmospheric simulation-based dataset, which can be used for assisting interpolation of in-situ data, especially in regions where rain gauges are sparse. In addition, some studies observed remarkable seasonal variations of precipitation gradient (Li and Fu, 1984; Putkonen, 2004; Wulf et al., 2010; Zhao et al., 2011), which implies that precipitation gradient can be related to weather conditions. However, very limited works have been done to investigate their relationships. Therefore, this study also investigates the relations between precipitation gradients and two meteorological factors (i.e. humidity and wind speed), to explore whether these factors can provide potential auxiliary information for adjusting the precipitation gradient in a region.

## 2. Description of datasets

### 2.1. Precipitation datasets

The dataset used to quantify the precipitation gradients is produced by Jiang et al. (2021). It covers

the whole TP and is generated by combing the ERA5 reanalysis (Hersbach et al., 2021) with the high-resolution simulated precipitation produced by Zhou et al. (2021). Three main steps are involved

to produce this dataset. First, a short-term high-resolution WRF (Skamarock et al., 2008) simulation with a horizontal resolution of 1/30° is conducted and the short-term simulation covers two representative years (2013 and 2018). Second, the precipitation from ERA5 is corrected with the high-resolution simulated precipitation using a CDF (Cumulative Distribution Function) matching method. Third, the high-resolution simulation is resampled to the spatial resolution of ERA5 and a

convolutional neural network-based model is trained using the resampled precipitation as the input and the original high-resolution simulated precipitation as the target, and then the corrected ERA5 precipitation is downscaled to a resolution of 1/30° using the trained model. This downscaled precipitation shows similar performance in describing the spatial variability of precipitation to the WRF simulations produced by Zhou et al. (2021), while it has a wide temporal coverage spanning 39 years

from 1980 to 2018, which allows us to investigate the interannual variations of precipitation gradients in the TP. For convenience, this downscaled precipitation is called ERA5_CNN hereafter. Previous evaluation of this dataset showed that it is skillful in reflecting spatial variability of precipitation and its bias has been much reduced compared with other analysis data. Nevertheless, it still overestimates the precipitation amount in the TP (Jiang et al., 2021).

This study also investigates the performance of IMERG (Integrated Multi-satellite Retrievals for Global Precipitation Measurement; Huffman et al., 2019) and HAR V2 (High Asia Refined Analysis version 2; X. Wang et al., 2020) in reflecting altitude dependence of precipitation, and compares it with that from ERA5_CNN. IMERG is the latest generation of global satellite-based precipitation products. The final run version of IMERG V06 with a horizontal resolution of 0.1° is used in this study, which

has applied gauge observations to correct the satellite estimates. HAR V2 is produced by dynamically downscaling the ERA5 reanalysis using the WRF model. It also covers the whole TP but has a coarser horizontal resolution (10 km) than ERA5_CNN.

    Observations from six rain gauge networks are used in this study. Five rain gauge networks with relatively high density but covering small sub-regions of the TP are used to validate the altitude

dependence of precipitation in these gridded datasets at small spatial scales. Details about the five rain gauge networks are given in Table 1 and their distributions are shown in Fig. 1. Besides, the network from the CMA (China Meteorological Administration) is also used in this study, which covers a large area of the TP but has scarce gauge density. Therefore, this network is used for quantifying the bias in ERA5_CNN in the TP.


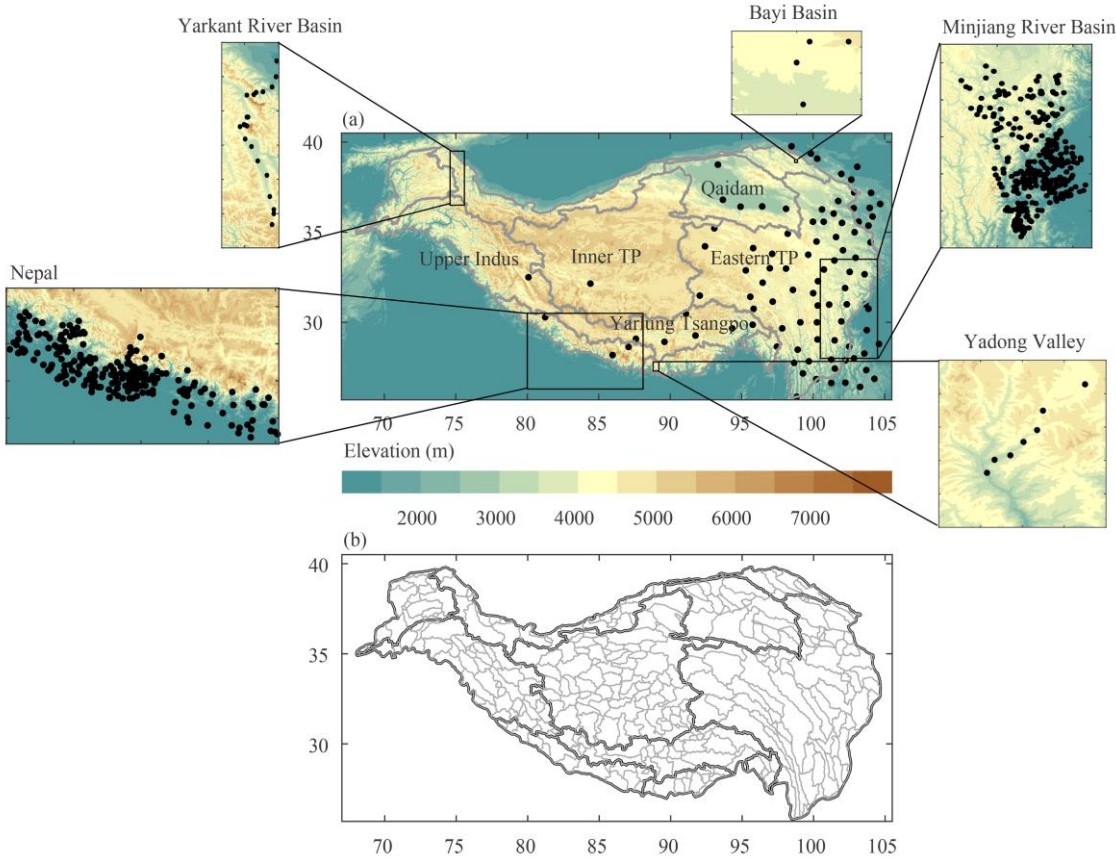

**Figure 1** (a) Topography of the Third Pole (TP) region and the boundaries of five sub-regions of the TP, along with the distribution of the five rain gauge networks. The black points represent the rain gauges. (b) The boundary of the 388 sub-basins in the TP. Fig. 1b shows the area above 2500 m a.s.l. The boundaries of the TP and the five sub-regions are derived from Zhang (2019).


**Table 1** Basic information about the six rain gauge networks used in this study.

| Rain gauge network | Temporal coverage | Number of gauges | Source |
|---|---|---|---|
| Yarkant River Basin | 2014.01-2015.12 | 28 | (Kan et al., 2018) |
| Bayi Basin | 2018.07-2018.09 | 4 | (Han et al., 2020) |
| Mingjiang River Basin | 2017.01-2017.12 | 375 | The Hydrological Bureau of the Ministry of Water Resources (MWR) in China |
| Yadong valley | 2018.07-2018.09 | 9 | (Yang, 2020) |
| Nepal | 2014.01-2016.12 | 283 | The Department of Hydrology and Meteorology (DHM) in Nepal |
| Whole TP | 1980.01-2018.12 | 95 | The China Meteorological Administration |

## 2.2. Other datasets

The elevation data used in this study is from the NASA Shuttle Radar Topographic Mission (SRTM), which provides global digital elevation data (DEM) at a resolution of 90 m. The 90-m DEM is resampled to 1/30° by averaging the elevation of all 90-m grids within a 1/30° grid to match the horizontal resolution of the precipitation data.

The ERA5 reanalysis data of near-surface humidity and wind speed are also used to explore the relations between precipitation gradients and meteorological factors.

## 3. Method

The precipitation gradients are calculated based on a linear regression between precipitation and altitudes, which can be expressed as follow:

$$P = a \times H + b, \tag{1}$$

Where $P$ is average precipitation (unit in mm.day$^{-1}$) for a specific period, $H$ is altitude (unit in 100 m), $a$ is the absolute precipitation gradient (APG; unit in mm.day$^{-1}$.100 m$^{-1}$) within a specific region, and $b$ is the intercept of the regression equation. In this study, the regression equation is fitted in 388 sub-basins of the TP above the 2500 m a.s.l. contour. The geometries of the 388 sub-basins (shown in Fig. 1b) are obtained from the HydroATLAS database (Linke et al., 2019), which provides twelve nested levels of sub-basins for the global. The level 6 sub-basins are applied in this study and these sub-basins have areas ranging from 2.91 km$^2$ to 120135.00 km$^2$. The relatively small size of these sub-basins can ensure that the grids used to fit the equations are dominated by similar prevailing winds. Moreover, the basin-scale precipitation gradient is easier to be applied for hydrological applications than the gridded precipitation gradient. The value of precipitation gradient for a sub-basin is given only when the following three principles are met: (1) the number of grids within the sub-basin should not be less than 10; (2) the standard deviation of altitude within the sub-basin should not be less than 50 m; (3) the $p$-value of the Student's t-test for the regression equation should be less than 0.05.

Although the ERA5_CNN shows good performance in representing spatial variability of precipitation, it has a systematic bias in the TP (Jiang et al., 2021). Therefore, the relative precipitation gradient (RPG, unit in %.100 m$^{-1}$) is also presented in this study. The RPG is calculated as follows:

$$RPG = \frac{a}{\bar{P}} \times 100\%, \tag{2}$$

Where $a$ is the absolute precipitation gradient from Equation (1) and $\bar{P}$ is the basin-average

precipitation. For calculating the RPG, $\overline{P}$ should be greater than 0.1 mm.day$^{-1}$.

To quantify the bias of ERA5_CNN, both absolute bias (Abias) and relative bias (Rbias) are used in this study, calculated as follows.

$$Abias = \frac{1}{n}\sum_{i=1}^{n}(M_i - O_i) , \qquad (3)$$

$$Rbias = \frac{\sum_{i=1}^{n}(M_i - O_i)}{\sum_{i=1}^{n}M_i} , \qquad (4)$$

Where $M_i$ and $O_i$ are the precipitation from ERA5_CNN and in-situ data, respectively, and $n$ is the number of samples.

Besides, the coefficient of variation (CV) is used to quantify the spatiotemporal variability of variables, which is defined as follow in this study.

$$CV = \frac{\sigma}{|\mu|} , \qquad (5)$$

Where $\sigma$ and $|\mu|$ are the standard deviation and absolute mean of a series of samples, respectively. The CV is dimensionless. The closer the CV value is to zero, the smaller the dispersion is.

## 4. Results

### 4.1. Validation of the altitude dependence of precipitation

The altitude dependence of precipitation from ERA5_CNN is compared with that from rain gauge data in the five networks mentioned in section 2.1. For comparison, the altitude dependence of precipitation from two widely-used precipitation datasets, i.e. IMERG and HAR V2, are also investigated.

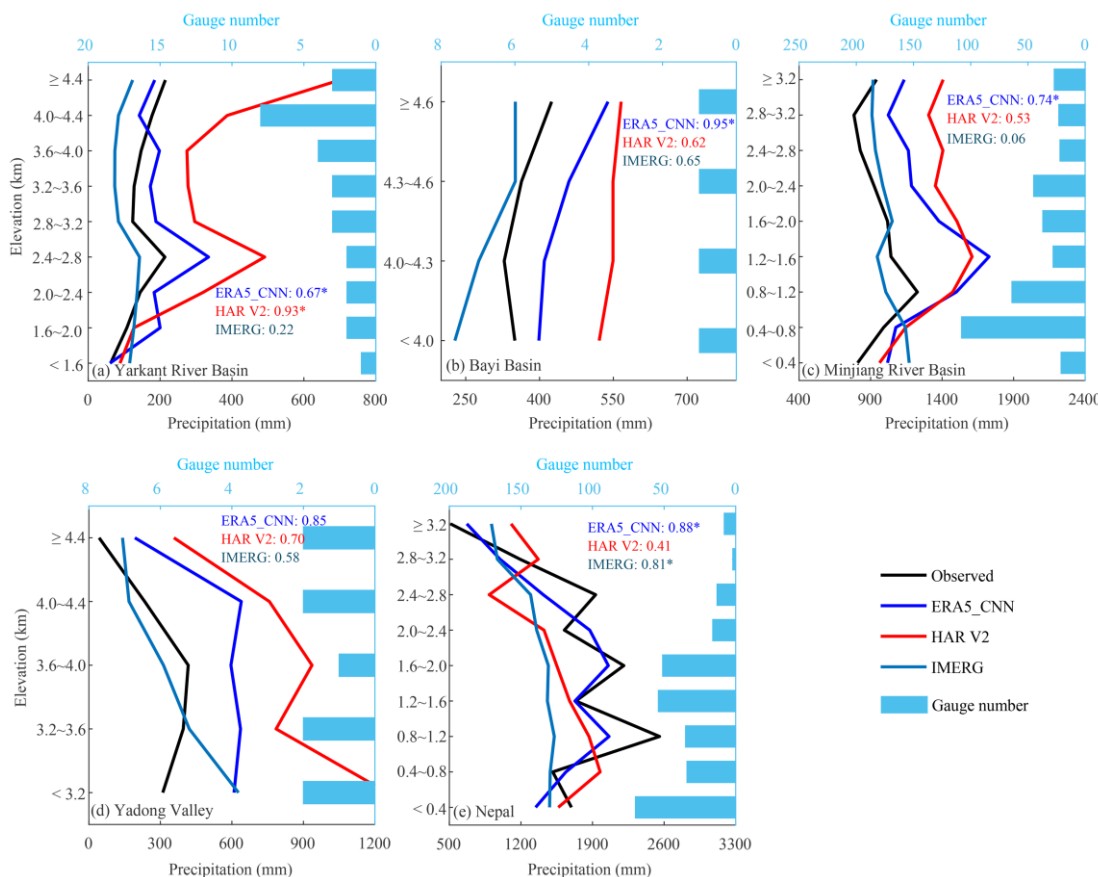

**Figure 2** Comparison between the altitude dependence of precipitation from ERA5_CNN, IMERG and HAR V2 and that from rain gauge data in five networks. The lines show the average precipitation amount in each altitude zone and the bars denote the number of rain gauges in each zone. The numbers in the figures give the spatial correlations of precipitation amount between the rain gauge data and precipitation products. The '*' represents the correlation is significant at the 95%

confidence level.

In the Yarkant River basin (Fig. 2a), all datasets reproduce the observed local precipitation maxima at 2400-2800 m a.s.l. Nevertheless, remarkable differences exist in these datasets. HAR V2 has the highest spatial correlation (0.93) with rain gauge data, but presents a sharp precipitation gradient above 4000 m a.s.l. ERA5_CNN also shows similar altitude dependence of precipitation to gauge data

but yields another local precipitation maximum at 1600-2000 m a.s.l., leading to a smaller correlation of 0.67. IMERG slowly changes with altitude with the lowest correlation of 0.22.

In the Bayi Basin (Fig. 2b), ERA5_CNN shows the most consistent pattern with rain gauge data with a correlation of 0.95, although it generally overestimates precipitation. In terms of the other two datasets, precipitation from IMERG decreases with altitude above 4600 m a.s.l., while precipitation

from HAR V2 has a similar magnitude at all altitudes.

In the Minjiang River Basin, Fig. 2c shows that precipitation from rain gauge data increases with

altitude below 1200 m a.s.l., then decreases with altitude between 1200-3200 m a.s.l., then rises again above 3200 m a.s.l. ERA5_CNN overestimates precipitation in this basin, but it shows the most similar altitude dependence to rain gauge data and has the highest correlation of 0.74. HAR V2 also generally reproduces the observed pattern but changes slowly with altitude above 1600 m a.s.l, yielding a smaller correlation of 0.53. However, precipitation from IMERG changes little with altitudes in the Minjiang River Basin.

In the Yadong Valley (Fig. 2d), ERA5_CNN firstly increases slowly and then decreases sharply with altitude and has a spatial correlation of 0.85 with gauge data, although it has a higher altitude of precipitation maximum than rain gauge data. Precipitation from HAR V2 also shows a similar pattern to that of rain gauge data above 3600 m a.s.l., but a different pattern below 3600 m a.s.l. In contrast, The altitude dependence of precipitation from IMERG is opposite to the observed one in the Yadong Valley.

In Nepal (Fig. 2e), precipitation amount from rain gauge data shows large fluctuation among different altitude bands. Generally, it increases with altitude below 2000 m a.s.l. and then decreases beyond this altitude level. It can be found that ERA5_CNN can better represent the altitude dependence of observed precipitation than the other two products (the spatial correlations with gauge data for ERA5_CNN, HAR V2 and IMERG are 0.88, 0.41 and 0.81, respectively), particularly in reproducing the great fluctuation of precipitation.

Overall, the high-resolution atmospheric simulation-based ERA5_CNN can reasonably represent the altitude dependence of precipitation in the TP and generally shows better performance than the widely-used IMERG and HAR V2. Therefore, it is used to quantify the spatial and temporal variability of RPGs in the TP.

## 4.2. Spatial patterns of precipitation gradients

Figure 3a shows the spatial distribution of the correlations between annual average precipitation from 1980 to 2018 and altitudes in each sub-basin. As shown in Fig. 3a, there are strong correlations between precipitation and altitude in many sub-basins with absolute correlations larger than 0.50 at about 55% of the sub-basins, and the correlations are significant at the 95% confidence level at most sub-basins. Therefore, it is feasible to interpolate precipitation based on precipitation gradients in the TP.

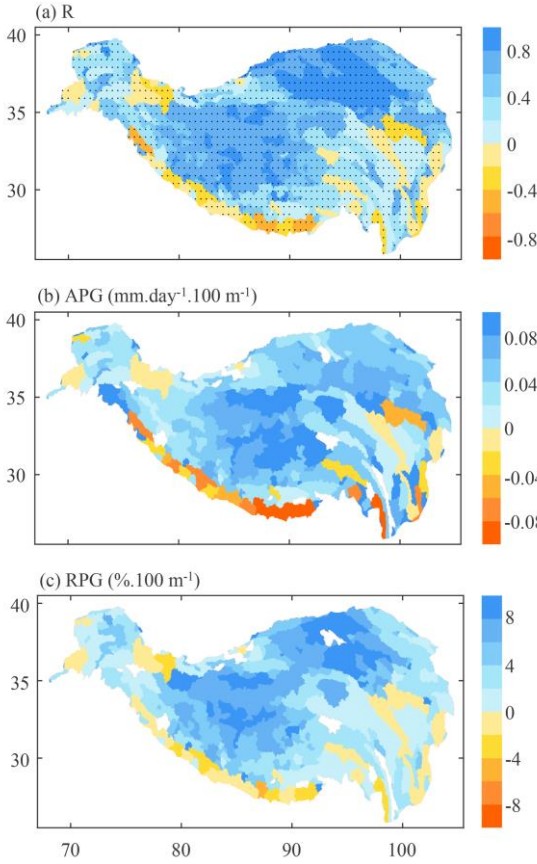

**Figure 3** Spatial distribution of (a) correlations between the annual average precipitation and altitude for all grids within each basin, (b) absolute precipitation gradients (APGs, precipitation change per 100 m altitude difference) and (c) relative precipitation gradients (RPGs, APGs divided by basin-average precipitation). The APGs and RPGs are calculated based on annual precipitation averaged from 1980 to 2018. The dots in Fig. 3a represent the correlations significant at the 95% confidence level. In Fig. 3b and c, the sub-basins with weak relationships between precipitation and altitude or no data value of RPG are filled with white.

Generally, at the annual scale, most sub-basins (about 81% of the total) in the TP have a positive APG and the sub-basins with negative APG are mainly distributed in the Himalayas, the Hengduan mountains in the eastern edge of the TP and the Western Kunlun in the northwestern TP, resulting in an average APG across all sub-basins of 0.05 mm.day$^{-1}$.100 m$^{-1}$. As shown in Fig. 3b and Table 2, large positive APG mainly occurs in the Inner TP, with an average value of 0.07 mm.day$^{-1}$.100 m$^{-1}$, followed by the eastern TP (covering the Yellow, the Yangtze, the Lancang and the Nu River Basin) (0.06 mm.day$^{-1}$.100 m$^{-1}$), the Yarlung Tsangpo River Basin and the Qaidam Basin (0.05 mm.day$^{-1}$.100 m$^{-1}$), and the upper Indus has the smallest APG of 0.04 mm.day$^{-1}$.100 m$^{-1}$. In some specific regions, such as the Qilian Mountains (Wang et al., 2009; Han et al., 2020) and some small basins in the southern TP (Li Wang et al., 2018; Zeng et al., 2021; Zhang et al., 2015) where the observed precipitation generally

increases with altitude, our study reports consistent results. Notably, most sub-basins along the Himalayas show large negative APG. This is also consistent with previous studies (Andermann et al., 2011; Bookhagen and Burbank, 2006; Chen et al., 2020; Salerno et al., 2015; Tang et al., 2018), which have demonstrated that there is a shape decrease in precipitation above 2500 m a.s.l. in this region.

As shown in Fig. 3c, the annual RPG generally has a similar spatial pattern to APG but shows large values in the Qaidam Basin. The average RPG across all the sub-basins of the TP is 4.25 %.100 m$^{-1}$. However, the RPGs show great spatial variability, ranging from -5.23 %.100 m$^{-1}$ to more than 20.00 %.100 m$^{-1}$. Quantitatively, the average RPGs within five sub-regions of the TP are shown in Table 2. The Qaidam Basin has the largest value of 11.26 %.100 m$^{-1}$, followed by the Inner TP with a value of 7.08 %.100 m$^{-1}$, and then the Upper Indus with a value of 3.17 %.100 m$^{-1}$. The Yarlung Tsangpo River Basin and the eastern TP have the RPG of 3.00 %.100 m$^{-1}$ and 2.90 %.100 m$^{-1}$, respectively. Generally, the spatial pattern of RPG shown in our study is in agreement with the result of Guo et al. (2016), which pointed out that large precipitation gradients are mainly in the Qaidam Basin but small in the Hengduan Mountains in the southeastern TP.

**Table 2** The APG and RPG averaged across the sub-basins within the five sub-regions and the whole TP with respect to different seasons. ETP: eastern TP; YTR: Yarlung Tsangpo River Basin; ITP: Inner TP; QDM: Qaidam Basin; UID: Upper Indus

| | ETP | YTR | ITP | QDM | UID | TP |
|---|---|---|---|---|---|---|
| APG (mm.day$^{-1}$.100 m$^{-1}$) | | | | | | |
| Winter | 0.01 | 0.01 | 0.01 | 0.01 | 0.04 | 0.02 |
| Spring | 0.04 | 0.02 | 0.04 | 0.04 | 0.04 | 0.04 |
| Summer | 0.16 | 0.13 | 0.19 | 0.10 | 0.05 | 0.11 |
| Autumn | 0.05 | 0.03 | 0.05 | 0.03 | 0.02 | 0.04 |
| Annual | 0.06 | 0.05 | 0.07 | 0.05 | 0.04 | 0.05 |
| RPG (%. 100 m$^{-1}$) | | | | | | |
| Winter | 4.01 | 6.04 | 13.20 | 7.53 | 3.65 | 5.06 |
| Spring | 2.76 | 6.04 | 8.99 | 11.69 | 3.90 | 5.11 |
| Summer | 3.21 | 2.87 | 6.67 | 10.47 | 2.81 | 4.20 |
| Autumn | 2.42 | 3.51 | 7.39 | 11.37 | 3.02 | 4.33 |
| Annual | 2.90 | 3.00 | 7.08 | 11.26 | 3.17 | 4.25 |

**4.3 Temporal variation of precipitation gradients**

**4.3.1 Seasonal patterns**

The APG and RPG at each basin are also calculated based on the seasonal average precipitation, and presented in Fig. 4 and Fig. 5, to explore the seasonality of the precipitation gradient.

As shown in Fig. 4 and Table 2, the absolute values of APG in summer are remarkably larger than those in other seasons, with an averaged value across all sub-basins in the TP of 0.11 mm.day$^{-1}$.100 m$^{-1}$, while they are 0.02 mm.day$^{-1}$.100 m$^{-1}$ in winter, and 0.04 mm.day$^{-1}$.100 m$^{-1}$ in spring and autumn. Such a seasonal pattern is not surprising because a large precipitation amount tends to result in a large value of APG and vice versa. Figure 5 shows the spatial distribution of RPG in the four seasons. In winter,

precipitation in some sub-basins is very small, therefore, the RPG is not calculated in these basins and masked with white. Unlike APG, RPGs in spring and autumn are larger than those in summer, which is especially true in the central TP. In winter, although many sub-basins are masked as no data, most of the remained sub-basins have the largest RPG among the four seasons. Table 2 shows that the values of RPG averaged across all the sub-basins in the TP are 5.06 %.100 m$^{-1}$ in winter, 5.11 %.100 m$^{-1}$ in

spring, 4.20 %.100 m$^{-1}$ in summer and 4.33 %.100 m$^{-1}$ in autumn.

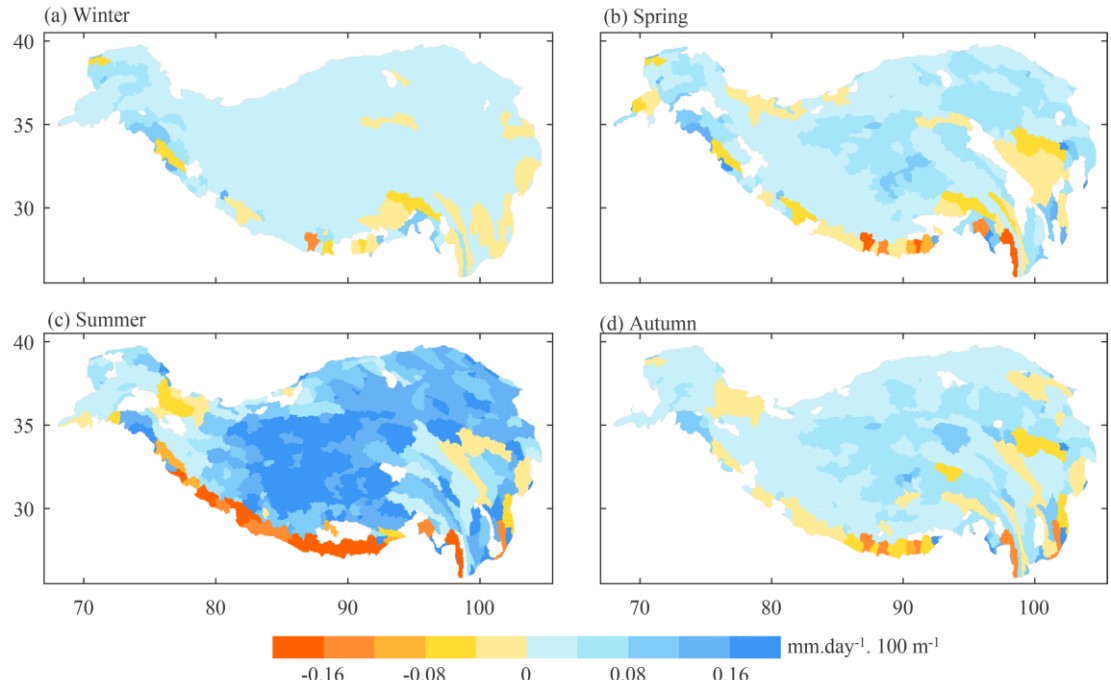

**Figure 4** Spatial distribution of APGs in (a) winter (December to February), (b) spring (March to May), (c) summer (June to August) and (d) autumn (September to November). The APGs are calculated based on seasonal precipitation averaged from 1980 to 2018. The sub-basins with weak

relationships between precipitation and altitude are filled with white.

Particularly, remarkable seasonal variation of precipitation gradient (both APG and RPG) can be found in the Himalayas. In winter, most of the sub-basins in this region have a positive precipitation gradient, however, it can be seen from Fig. 4c and 5c that this region is dominated by negative

gradients in summer. In spring (Fig. 4b and 5b) and autumn (Fig. 4d and 5d), the western Himalayas

has a positive gradient and the eastern Himalayas has a negative gradient. This phenomenon was also

observed by Wulf et al. (2010) who found that in the northwest Himalayas the precipitation gradients

are reversed between winter and summer, as well as by Putkonen (2004) who reported that in the Nepal

Himalayas monsoon precipitation maximum occurs at the altitude of about 3000 m a.s.l., while

precipitation continuously increases with altitude in dry seasons.

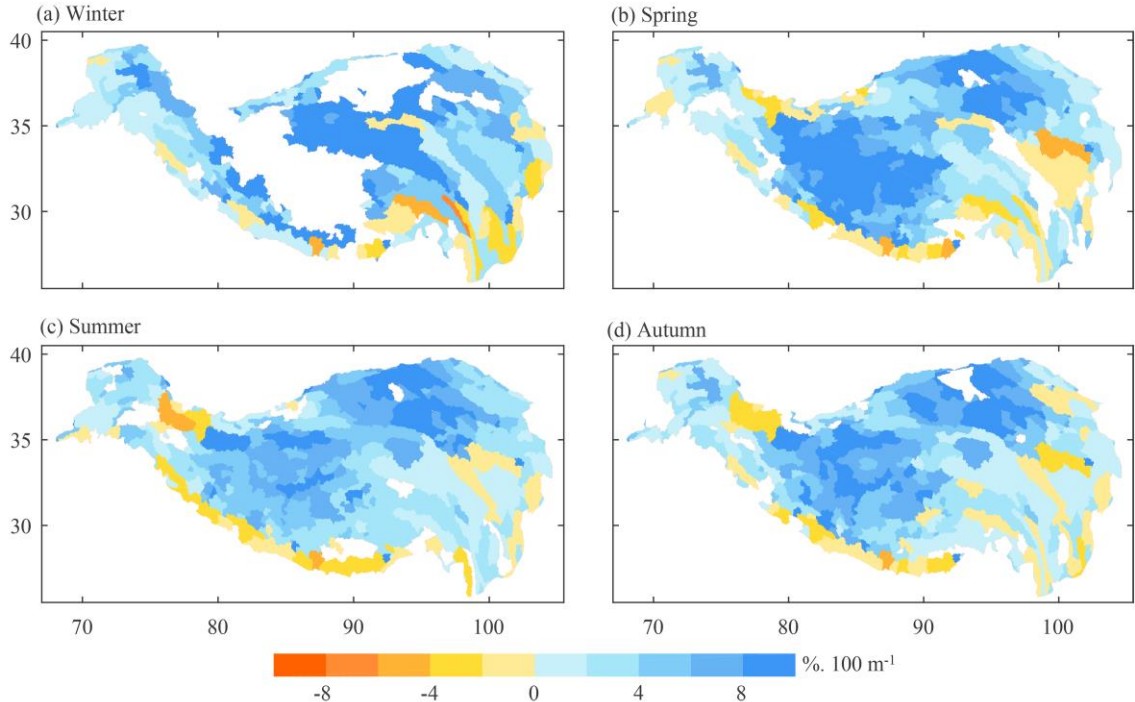


**Figure 5** Same as Fig. 4 but for RPGs. The sub-basins with weak relationships between precipitation

and altitude or no data value of RPG are filled with white.

In summary, the precipitation gradient (including both APG and RPG) in the TP shows great

seasonal variation and it may be desirable to interpolate seasonal or monthly precipitation with

precipitation gradients calculated at the corresponding season or month, especially in the Himalayas.

**4.3.2 Interannual variations**

The CV and trend for annual APG and RPG during 1980-2018 are calculated for each sub-basin of

the TP. As shown in Fig. 6a and 6b, the CV values for both APG and RPG at most sub-basins are less

than 0.2, implying low inter-annual variability of precipitation gradient. Fig. 6c shows that APG has a

positive trend at most sub-basins, especially in the Inner TP, which was also reported by Guo et al.

(2016) who used in-situ data to characterize the precipitation gradients in the TP. Such a pattern of

precipitation gradient trend is mainly because the TP has overall become wetter in recent decades,

especially in the central and northern TP (Sun et al., 2020; X. Wang et al., 2018; Yang et al., 2014). In contrast, RPG does not show a positive trend at most sub-basins, and its trend in fewer basins is significant at the 95% confidence level. The average value of the RPG trend across all sub-basins is -0.0042 %.100 m$^{-1}$.year$^{-1}$. Therefore, RPG is less sensitive to the climatic change of precipitation amount, and the RPG obtained in a certain period is expected to be more representative than APG when applying for precipitation interpolation under climate change.

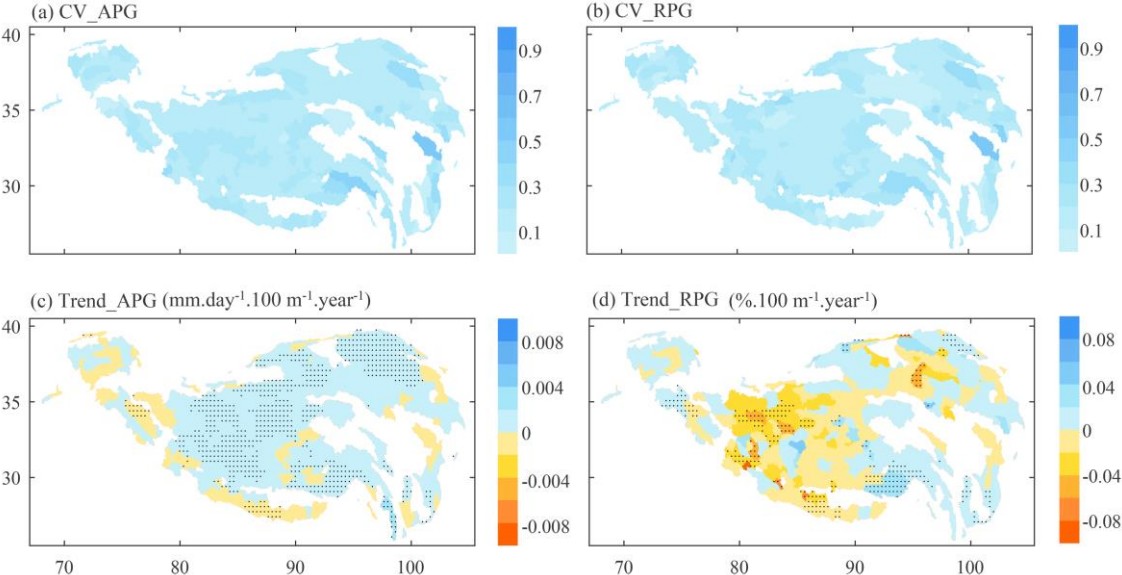

**Figure 6** Spatial distribution of (a) and (b) the coefficient of variation (CV) and (c) and (d) trend for annual APGs and RPGs during 1980 to 2018. The dots in Fig. 6c and d represent trend significant at the 95% confidence level. The CV and trend are calculated only for sub-basins without missing APG or RPG during 1980-2018.

## 5. Discussions

### 5.1 Uncertainties in APGs and RPGs

This study uses the atmospheric simulation-based ERA5_CNN to characterize the precipitation gradients in the TP. However, the ERA5_CNN has biases in the TP, leading to uncertainties in the calculated APGs and RPGs. Figure 7 shows the Abias and Rbias of annual precipitation from ERA5_CNN during 1980-2018 at the locations of CMA stations. It can be found that ERA5_CNN generally overestimates precipitation in the TP with Abias ranging from -365.99 mm.year$^{-1}$ to more than 1500.00 mm.year$^{-1}$ and Rbias ranging from about -30.00% to more than 150.00%. This result is similar to previous works that demonstrated overall wet bias in atmospheric simulation in the TP (Gao et al., 2015; Y. Wang et al., 2020; Zhou et al., 2021). According to the definition of APG and RPG, if

Abias is spatially homogeneous (i.e. the ERA5_CNN has the same absolute value of overestimation or underestimation at all locations in a region), the slopes of the regression line derived from ERA5_CNN and rain gauge data are the same because these two lines are parallel (as shown in Fig. S1a); if Rbias is uniform in space (i.e. the ERA5_CNN has the same percentage of overestimation or underestimation), the calculated RPG is consistent with that from rain gauge data because both APG and the basin-average precipitation in Equation 2 have the same percentage of bias, which is also illustrated in Fig. S1b. By comparing Fig. 7a and 7b, we can find that Rbias is more homogeneous than Abias (the CV value for all CMA stations is 1.24 for the Rbias, while it is 1.59 for the Abias). In this case, it is expected that RPG derived from the ERA5_CNN is closer to the observed one than APG and more appropriate for interpolating rain gauge data.

Nevertheless, the Rbias still has great spatial variability. Given the complexity of biases in ERA5_CNN, we recommend comparing the APG and the RPG and selecting the better one for specific applications.

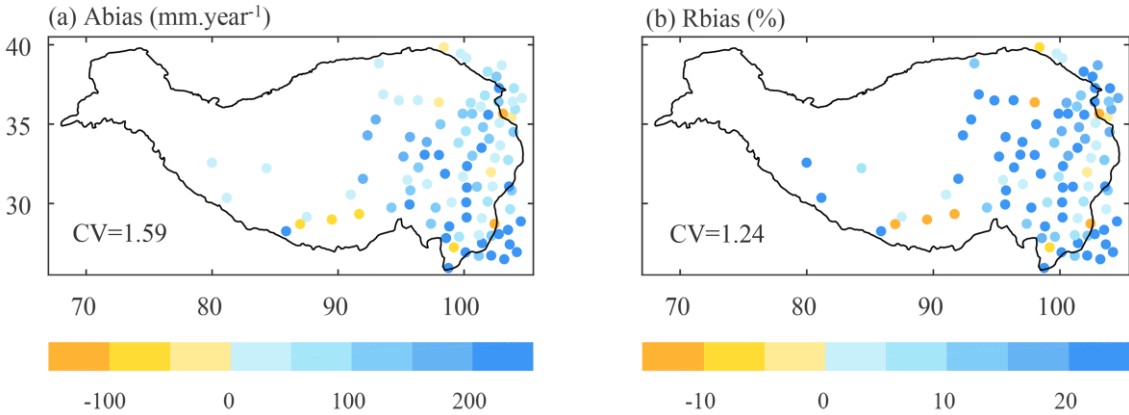

**Figure 7** Spatial distribution of (a) absolute bias (Abias) and (b) relative bias (Rbias) for annual precipitation from ERA5_CNN during 1980-2018 at stations of China Meteorological Administration.

**5.2 Relations between precipitation gradients and relative humidity and wind speed**

Previous works mainly focused on the influence of static topographic parameters (e.g. altitude, slope, aspect and exposure) on precipitation gradient (Basist et al., 1994; Diodato, 2005; Sevruk, 1997; Singh et al., 1995). However, section 4 shows that precipitation gradients are likely related to meteorological conditions. Therefore, this section discusses the possible factors that may influence the spatiotemporal variations of precipitation gradient. The near-surface relative humidity and wind speed are selected as the potential factors because they should be the indicators of mass and dynamic conditions for the formation of precipitation, respectively. Given that the magnitude of APG is likely to

be influenced by precipitation amount (as shown in Fig. 4) and the RPG is more informative, this
section only discusses the relations between the RPG and the two meteorological factors.

Our results show that large RPG mainly occurs in the Qaidam basin and Inner TP characterized by dry air conditions. In addition, RPG has larger values in winter and spring than in summer. Similar results have been reported in the Himalayas (Putkonen, 2004), the Xinjiang region (Zhao et al., 2011) and the Qinling Mountains (Li and Fu, 1984), which found that the altitude with precipitation maximum in dry seasons is higher than that in wet seasons. These results indicate that there may be a close relationship between RPG and the humidity of air mass. Therefore, the relationships between annual average relative humidity and annual RPG are investigated in the whole TP and its five sub-regions. As shown in Fig. 8, there is a good linear relationship (R=-0.68) between relative humidity and RPG when considering all the sub-basins in the TP (Fig. 8a). In terms of each sub-region, the negative correlation between relative humidity and RPG is relatively small in the Yarlung Tsangpo River Basin (Fig. 8c) and the Inner TP (Fig. 8d), while they are larger than 0.5 in the other three sub-regions (Figs. 8b, e and f). Overall, the RPG generally decreases with increasing relative humidity in all sub-regions, indicating that precipitation tends to occur at lower altitudes when the relative humidity is larger, which is easy to understand because air masses with lower humidity tend to be saturated after a higher uplift.

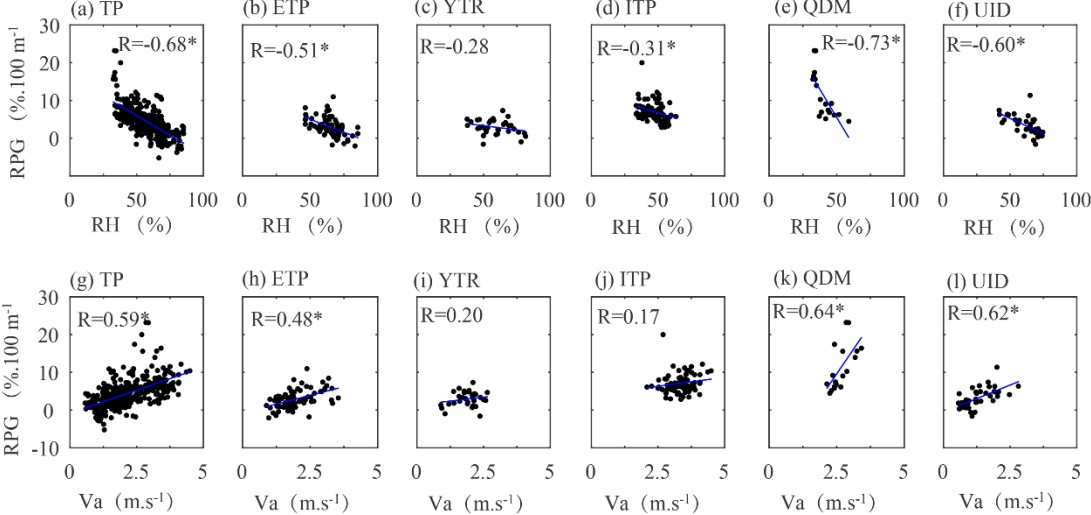

**Figure 8** Relationships between annual RPGs and (a-f) basin-average relative humidity (RH) and (g-l) wind speed (Va) in different sub-regions of the TP. ETP: eastern TP; YTR: Yarlung Tsangpo River Basin; ITP: Inner TP; QDM: Qaidam Basin; UID: Upper Indus

The relations between near-surface wind speed and RPG were also tested (Fig. 8g-l). It can be found that three sub-regions have high positive correlations (R>0.45) between RPG and wind speed, and the Yarlung Tsangpo River Basin and the Inner TP have lower positive correlations (R=0.20 and

0.17, respectively). The correlation coefficient for the whole TP is as high as 0.59, indicating RPG increases with increasing wind speed. The positive correlations between precipitation gradient and

380 wind speed reported in this study have also been demonstrated in previous studies, e.g. Johansson and Chen (2003) found that precipitation in Sweden increases with increasing wind speed on the upwind side of mountains; Hill (1983) also confirmed that wind direction and wind speed could have great impacts on the distribution of precipitation enhancement in mountainous regions. Moist air blocked by upwind barriers usually leads to enhanced precipitation in the windward slopes, which is one of the

385 main mechanisms of orographic precipitation in mountainous regions (Houze, 2012; Roe, 2005). Thus, the strong wind tends to bring more moisture to high altitudes and further results in precipitation enhancement in high altitudes. That is why strong wind tends to result in larger positive precipitation gradients.

**5.3 Impact of spatial scale on the estimation of precipitation gradients**

In this study, the precipitation gradient is fitted using all grids with a specific sub-basin, therefore, the estimated precipitation gradient is likely to be spatial scale-dependent. Accordingly, we investigate and compare precipitation gradients at different spatial scales by calculating the precipitation gradients based on four sub-basin levels provided by the HydroATLAS database. Figure 9 shows the spatial distribution of the annual RPG calculated at different sub-basin levels (a lower sub-basin level has a

larger spatial scale). The results of APG are similar to those of RPG and thus not shown. It can be seen that RPGs calculated at different spatial scales differ greatly, especially in the southern and eastern TP where the topography is complex. For example, the RPGs are negative in the southern TP when calculated for large river basins (Fig. 9a), while they tend to be positive in sub-basins of these large river basins. This is similar to the results of Sun and Su (2020) who reported that precipitation overall

decreases with increasing altitude in the Yarlung Tsangpo River Basin but shows the opposite variation in some small sub-basins. Taking the TP as a whole, we can find that the values of RPG increase from L4 to L6 and remain relatively stable after L6 (Fig.10a). In addition, the correlations between precipitation and altitude tend to be larger at smaller spatial scales (Fig. 10b). This indicates that precipitation variations at large scales are more controlled by large-scale atmospheric circulations but at

small scales are more dependent on local topography.

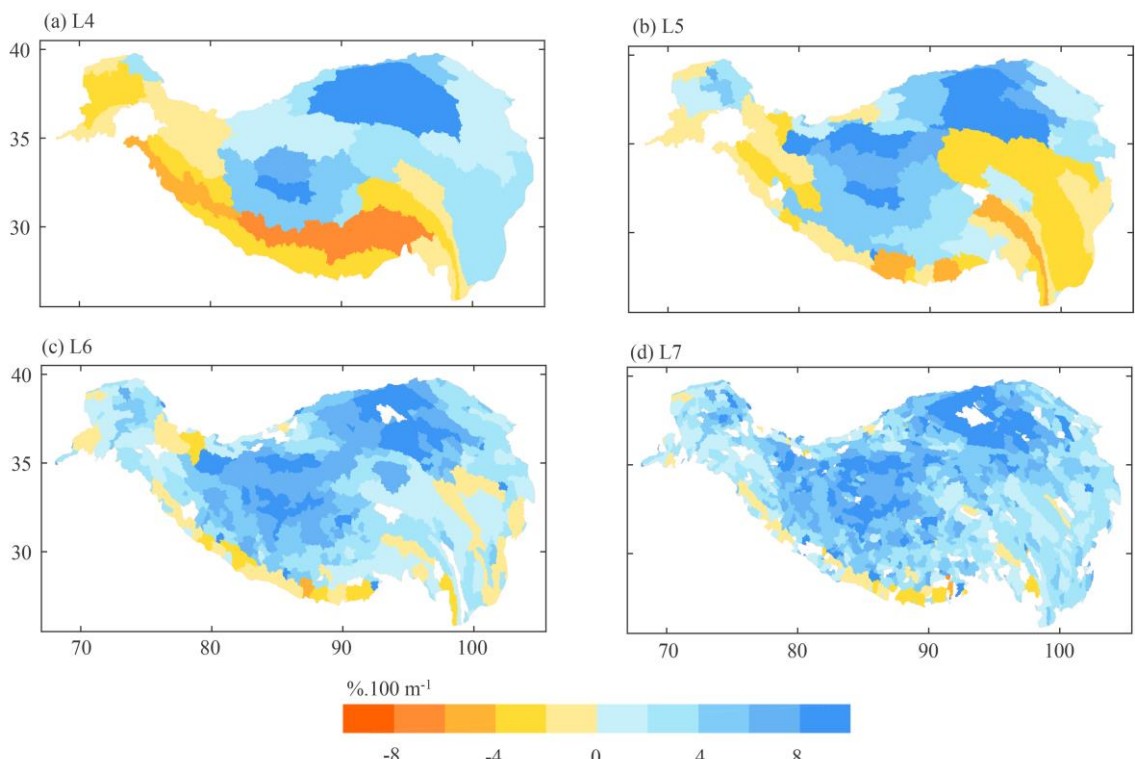

**Figure 9** Spatial distribution of annual RPGs calculated at four sub-basin levels. The spatial scales of sub-basins (i.e. sub-basin area) generally decrease from L4 to L7.

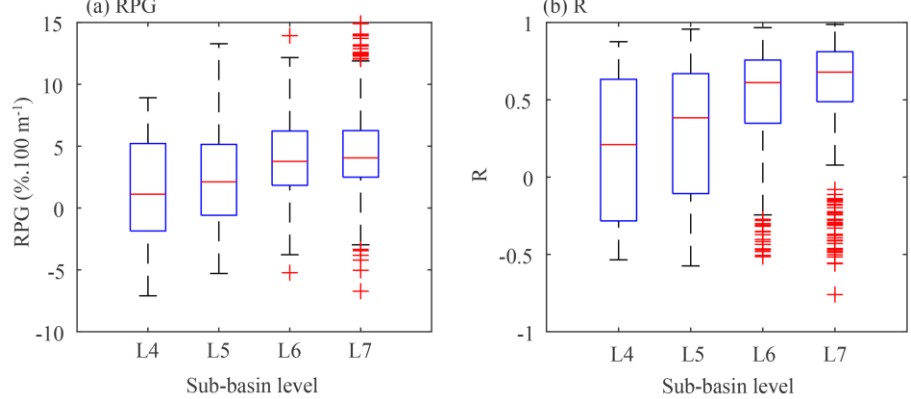

**Figure 10** Comparison of (a) RPGs and (b) the correlations between precipitation and altitude calculated at different sub-basin levels. Each box represents the distribution of RPGs or correlations of all the sub-basins over the TP. The red line shows the median value of RPGs. The bottom and top edges of the box represent 25th and 75th percentiles, respectively. The whiskers represent the extreme value. The red sign "+" shows the RPGs beyond the extreme value.

## 6. Conclusions

In this study, the altitudinal precipitation gradient in the TP is investigated at the basin scale using

a high-resolution atmospheric simulation-based precipitation dataset and its spatiotemporal variability is analyzed.

The performance of the high-resolution atmospheric simulation-based dataset in describing the altitude dependence of precipitation is firstly validated using observations from five rain gauge networks. The results show that this dataset can reasonably reproduce the observed altitude dependence of precipitation and generally performs better than the widely-used IMERG and HAR V2 in the TP.

Both absolute precipitation gradient (APG) and relative precipitation gradient (RPG) for annual and seasonal average precipitation are calculated for 388 sub-basins of the TP. Most sub-basins of the TP have positive precipitation gradients, and negative gradients are mainly distributed in the Himalayas, the Hengduan mountains and the Western Kunlun. The APGs are less than -0.05 mm.day$^{-1}$.100 m$^{-1}$ in the central and eastern Himalayas but greater than 0.06 mm.day$^{-1}$.100 m$^{-1}$ in most sub-basins of the central TP. Meanwhile, the annual RPGs range from about -5.00%.100 m$^{-1}$ in the Himalayas to more than 20.00%.100 m$^{-1}$ in the Qaidam Basin. Particularly, both APG and RPG show large values in the Inner TP. The seasonal variations of APG are corresponding to the seasonal variability of precipitation amount, with larger APG in wet seasons but smaller in dry seasons. However, the RPG has opposite seasonal variations.

The variations of precipitation gradient are related to meteorological conditions. Analyses show that the RPGs decrease with increasing relative humidity but increase with increasing wind speed. The relationships between RPGs and the two factors are strong with absolute correlations greater than 0.50 for both factors when taking all sub-basins in the TP into account. The strong correlations suggest that relative humidity and wind speed can be potential indicators to adjust RPG regionally.

In addition, our results show that the precipitation gradient in the TP is spatial scale-dependent. The precipitation gradients are positive in the northern TP but negative in the southern TP at larger spatial scales, however, they tend to be positive at smaller spatial scales, even in the southern TP. As the spatial scale decreases, the precipitation gradient first increases and then remains relatively stable. This is because the impact of large-scale atmospheric circulations on precipitation distribution is reduced and topography is the key determinant at a small scale, highlighting the importance of calculating precipitation gradient at a relatively small spatial scale.

In summary, our study presents the spatiotemporal variability of precipitation gradient in the TP, which can be used as a reference for assisting precipitation interpolation. Nevertheless, uncertainties still exist (as shown in section 5.1). Further works are expected to evaluate the accuracy of the obtained precipitation gradients in the TP, which requires reliable observations, e.g. high-quality radar observation or high-density rain gauge networks.

**Code and Data availability.** All codes used to produce the results are available upon request to the authors. The high-resolution atmospheric simulation-based precipitation dataset that supports this study is available upon request from the authors. Precipitation gradients (including both APG and RPG) for 388 sub-basins of the TP are provided in the supplementary file.

**Author contributions. Yaozhi Jiang**: Conceptualization, Data curation, Investigation, Formal analysis, Methodology, Software, Visualization, Writing – original draft preparation; **Kun Yang**: Conceptualization, Funding acquisition, Project administration, Resources, Funding acquisition, Supervision, Writing – review & editing; **Hua Yang**: Conceptualization, Data curation, Writing – review & editing; **Hui Lu**: Supervision, Writing – review & editing; **Yingying Chen**: Data curation,

Writing – review & editing; **Xu Zhou**: Methodology, Writing – review & editing; **Jing Sun**: Methodology, Writing – review & editing; **Yuan Yang and Yan Wang**: Writing – review & editing

**Conflict of interest.** The authors declare that they have no conflict of interest.

**Acknowledgments**. This work was supported by the Strategic Priority Research Program of Chinese Academy of Sciences (Grant No. XDA2006010201), National Science Foundation of China (Grant No.

41905087) and NSFC Basic Science Center for Tibetan Plateau Earth System (Grant No. 41988101). This is a contribution to CORDEX-FPS-CPTP. The dataset of the boundaries of the TP and its five sub-regions used in this study is provided by National Tibetan Plateau Data Center (http://data.tpdc.ac.cn)

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
