# Peer review of "Characterizing basin-scale precipitation gradients in the Third Pole region using a"

_Hydrology and Earth System Sciences, 2022_

## Author Comment (AC1)

**General response:** We sincerely thank the reviewer for the comments, which are helpful for us to improve the quality of this manuscript. The main concern of the reviewer is why the relative precipitation gradient (RPG) rather than the absolute precipitation gradient was presented. In the manuscript, the systematic bias of precipitation in ERA5_CNN is not well described, which may have misled the reviewer. **Actually, this "systematic bias" is the bias relative to the precipitation amount (in other words, its unit is percentage but not mm).** The justification is detailed below, which will be added to the coming revised manuscript. Brief responses to other comments are also listed below and further revisions will be given in the revised manuscript. We hope these responses have addressed all your concerns.

1. RPG problems

The definition of relative precipitation gradient (RPG) in Equation (2) is problematic from my perspective. There is no obvious linkage between precipitation intensity and precipitation gradient because precipitation gradient is just a spatial pattern no matter how large the mean value is (e.g.,1 0 or 1000). The authors state that RPG is used because ERA5_CNN has systematic bias in the TP. The most common systematic bias is overestimation or underestimation. If we assume the spatial distribution of overestimation or underestimation is even (cases with uneven distributions are more complex), the precipitation gradient has nothing to with the magnitude of precipitation, which means the normalization operation in Equation (2) is meaningless. Actually, the normalization could be harmful. For example, Let's assume a is 0.2 mm/m and P is 2 mm/day according to ERA5_CNN. Then, RPG=a/P=0.2/2 = 0.1. If we have another dataset which could have similar spatial distribution with ERA5_CNN but an overestimation bias. Let's say its P=4 mm/day. According to results from this study, the gradient for this new dataset will be 0.1 X 4 =0.4, which is incorrect. I can imagine that this simple example could be quite common in applications motivated by this work. The RPG problem could also result in wrong comparison between different products. For example, when comparing ERA5_CNN and gauge data in a basin, RPG cannot provide useful information. Even the RPG values of the two products are the same, their absolute precipitation gradients are often different because their mean precipitation is different. Similarly, even the RPG of a product is biased, its absolute precipitation gradient could be correct. Therefore, the use of RPG does not make sense.

Besides, I recommend the authors also present absolute precipitation gradient. RPG maps cannot show the impact of atmosphere vapor. For example, large RPG may

happen in dry basins, but its significance could be weaker compared to relatively smaller RPG in wet basins where precipitation amounts and bias are larger.

**Response**: Thanks for your comments! **The reviewer assumes that the systematic biases in precipitation dataset have the same absolute value in different altitudes or regions. In reality, we have assumed that the relative bias in precipitation is systematic.** We are sorry that the description about this assumption is not well clarified in the manuscript. In general, locations with large precipitation amounts tend to have large absolute values of bias and vice versa. For example, Gao et al. (2015, Evaluation of WRF mesoscale climate simulations over the Tibetan Plateau during 1979-2011. J. Clim. 28, 2823–2841) showed that the absolute values of bias are large in wet regions and small in dry regions of the Tibetan Plateau. This is also demonstrated in our evaluation results (Figure R1b). Accordingly, the relative bias is relatively uniform in different regions of the TP (Figure R1c). We will give more introduction about the systematic bias in ERA5_CNN in the revised manuscript.

[Figure]

**Figure R1** Distribution of (a) averaged annual precipitation (mm), (b) absolute bias (mm) and (c) relative bias (%) in ERA5_CNN during the period from 1980 to 2018.

Given that the relative biases are relatively uniform, it is appropriate to calculate the relative precipitation gradients (RPG). For example, there are two points, A (with actual precipitation amount of $P_1$ and altitude of $H_1$) and B (with actual precipitation amount of $P_2$ and altitude of $H_2$). Given that ERA5_CNN has the same relative bias of 10% at the two points, the precipitation amount in ERA5_CNN at A and B are $P_1 \times (1 + 10\%)$ and $P_2 \times (1 + 10\%)$, respectively.

The RPG derived from ERA5_CNN is

$$RPG = \frac{[P_2 \times (1+10\%) - P_1 \times (1+10\%)]/(H_2 - H_1)}{[P_2 \times (1+10\%) + P_1 \times (1+10\%)]/2} = \frac{(P_2 - P_1)/(H_2 - H_1)}{(P_2 + P_1)/2},$$

which is the **same** as that calculated from the actual precipitation.

Therefore, we presented the relative precipitation gradients in the manuscript.

2. Atmospheric model still has large bias in complex terrain including the study area which is also acknowledged in this study. This problem needs more discussion.

**Response**: Thanks for the comment! We will include more discussions about the uncertainties in atmospheric models in the revised manuscript.

3. The discussion between the relationship between precipitation and wind speed can be further improved. They are both aspects of air mass movement which is affected by both atmosphere and topography. If you try to investigate the causality (e.g., some sentences in abstract and discussion parts), you should go further. High or low wind speed is also the result of various environmental factors.

**Response**: This is a very good suggestion! We absolutely agree that orography has impacts on the spatial variability of both precipitation and wind speed. In the revised manuscript, we will discuss the relations between precipitation gradients and wind speed/relative humidity within different sub-regions of the TP that may have diverse orography.

4. Section 2.1: what's the time period of the 1/30-degree WRF simulations? Since ERA5_CNN is taken from Jiang et al. (2021), it is recommended to introduce more method and evaluation details in the manuscript.

**Response**: The 1/30-degree WRF simulations cover the whole years of 2013 and 2018. We will add more details about the production and evaluation of ERA5_CNN in the revised manuscript.

5. Section 2.2 introduces IMERG and HAR V2 which should be in "Section 2.1 Precipitation Datasets".

**Response**: Thanks for the suggestion. We will move them to Section 2.1.

6. Equation (1): units of P and H are needed.

**Response**: The units of P and H are "mm" and "100 m", respectively. We will add them in Equation (1).

7. Line 145-149: This description has a logical problem. It is stated that "the precipitation gradient is estimated only when the following three principles are met" and the third principle is "(3) the p-value of the Student's t-test for the regression equation should be less than 0.05". But you cannot know p-value without estimating precipitation gradient. Please revise this sentence.

**Response**: Thanks for pointing out this unclear description. Actually, we first fit the regression equation at each sub-basin, then, the Student's t-test for the regression equation is conducted. If the p-value of the Student's t-test is less than 0.05, the slope of the fitted line is treated as the precipitation gradient, otherwise, the precipitation of the sub-basin is filled with a missing value. We will revise this sentence as "The value of precipitation gradient for a sub-basin is given only when the following three principles are met: 1) the number of grids within the sub-basin should not be less than 10; 2) the standard deviation of altitude within the sub-basin should not be less than 50 m; 3) the p-value of the Student's t-test for the regression equation should be less than 0.05."

8. Figure 2: I suggest adding rain gauge number of different elevation bands.

**Response**: We will add the number of rain gauges in each elevation band in Figure 2.

9. Is Tibetan Plateau a more suitable word compared to "Third Pole"? Tibetan Plateau is clearer as a geographical concept.

**Response**: Our study region covers a wide region, including Pamir and Hindukush. This region is now called Third Pole.

10. The writing and presentation of the manuscript can be further improved.

**Response**: Thanks for the suggestion! We will thoroughly revise and edit the language

of the manuscript.

---

## Author Comment (AC2)

Thank you for the quick response. The responses to all comments look great to me, except for the RPG problem. I don't think the explanation of systematic bias can solve the problem. Before I state my reasons, I will just give my suggestion: the authors can include both relative and absolute precipitation gradients and thoroughly discuss absolute VS relative gradients in the manuscript. The current manuscript only has five figures presenting the quantitative results (Figures 2 to 6). I believe as a research article, it has enough room to include more results which will make this paper more interesting and informative. The comparison between absolute and relative gradients can partly solve the concerns, considering gradients from ERA5_CNN contain large uncertainties in the third pole.

**General response**: We sincerely thank the reviewer for spending lots of time on helping us improve the work. The suggestion including both absolute and relative precipitation gradients is pretty good, which can make this manuscript more informative. Therefore, we will present absolute precipitation gradients in the revised manuscript. In addition, we will introduce more details about the evaluations and biases of ERA5_CNN and discuss the uncertainties in both absolute and relative precipitation gradients in the revised manuscript to amplify our work. Responses to specific comments are detailed below and we will include more details during our revision.

The authors' explanation is that the systematic bias can be expressed as the relative bias (i.e., a fraction of precipitation amount), which is relatively uniform in different regions of the TP. Therefore, it is appropriate to calculate RPG. However,

1.  The experiments in my previous comments are not answered. RPG from different datasets/regions/periods is not comparable. For example, for cases where RPG1 from ERA5_CNN and RPG2 from rain gauge data are the same, we cannot say RPG1 is perfect or not because ERA5_CNN and RPG2 could have different mean precipitation. On the other hand, if RPG1 and RPG2 are different, it is still possible that ERA5_CNN captures the correct gradient pattern. Besides, the signs of RPG under/over estimation could be different from under/over estimation of absolute precipitation gradients, making the results-based RPG less reliable. Due to this problem, evaluation of ERA5_CNN using rain gauge data and comparing gradients in different regions of the third pole in the manuscript could be meaningless using RPG.

**Response:** As we have not fully understood this comment, here we clarify why we present RPG in this study in another way, which will be added as the background in the revised version.

It is common to interpolate precipitation in complex terrain with station data at lower elevations. In this case, the interpolation may be conducted with either PG or RPG. If both PG and RPG are accurate, the interpolation results should be the same. So, the question is: can RPG be estimated more easily than absolute PG? If RPG can be estimated more stably, then RPG is favorable; if absolute PG can be estimated more stably, then absolute PG is favorable. The biases in the precipitation mean of the dataset will propagate to the interpolation results when using the absolute precipitation gradients because the absolute precipitation gradients contain the biases from both mean and spatial variability of precipitation. As shown in Figure R1, HAR V2 with high precipitation amount generally has large absolute precipitation gradients. However, if two datasets have similar spatial variability but different means of precipitation (as ERA5_CNN and HAR V2 in Figure R1), they will have similar RPGs. Thus, using RPGs can partly eliminate the influence of biases in precipitation mean on interpolation results and is favorable.

In the previous comment, the reviewer gave an example that the same RPG value will lead

to different absolute gradients when datasets with different mean precipitation are used, accordingly, the reviewer said that RPG cannot provide useful information about the absolute precipitation variability. **In reality, the differences in absolute precipitation gradients were caused by the differences in precipitation mean, rather than RPG.**

[Figure]

**Figure R1** Spatial pattern of absolute precipitation gradients (PG; left panel) and relative precipitation gradients (RPGs; right panel) from ERA5_CNN, HAR V2 and IMERG. The precipitation gradients were calculated using average annual precipitation from 2008 to 2018.

2. There is no evidence that the relative bias is uniformly distributed in space. Relative bias is affected by many factors particularly in the large scale, while precipitation amount is just one of those factors. Actually, if relative bias can be so easily estimated, bias correction should be an easy task such as in the third pole, but the reality is that researchers are struggling with bias correction in complex terrain. I believe the authors hope that the RPG calculated in this study can be applied in other situations, but if the RPG is built on assumptions with large uncertainties, the application of RPG will be risky.

**Response:** We don't intend to say the results presented in this manuscript is the accurate one but we believe it is a forward step toward understanding the precipitation distribution in this region with complex terrain. As demonstrated in Figure R2, the value of Coefficient of Variation (CV, defined as the ratio of the standard deviation to the mean of a set of samples, which is used to measure the degree of dispersion of a set of samples) for relative biases over the TP is 1.2, while it is 1.6 for absolute biases (a smaller CV value means less dispersion). Therefore, the relative biases are

relatively uniform in space when compared with the absolute biases; in other words, using RPG can more reliably describe precipitation gradient. Nevertheless, we agree with you that the relative biases in ERA5_CNN vary in space and that biases in ERA5_CNN will result in uncertainties in RPG and will discuss the uncertainties in the revised manuscript.

[Figure]

**Figure R2** Distribution of (a) averaged annual precipitation (mm), (b) absolute bias (mm) and (c) relative bias (%) in ERA5_CNN during the period from 1980 to 2018.

3. The definition of bias is unclear. In evaluation studies, the relative bias is calculated against the reference dataset such as ground observations, but the calculation of RPG in this study is against the target dataset ERA5_CNN. I don't know how large the impact is, but this can weaken the reliability of RPG. For example, for a mountain slope, ERA5_CNN has low precipitation (P1) in low elevation and high precipitation (P2) in high elevation, I expect that P1 is more reliable than P2 because models are less reliable in high elevation. Using the method in this study, we can calculate RPG1 in low elevation and RPG2 in high elevation. Comparing the quality of RPG1 and RPG2 is cumbersome because we don't the direction (over or underestimation) of P1 and P2. Of course, this problem also affects absolute gradients, but after normalizing using P1 and P2, this problem becomes too complex.

**Response:** We need to clarify the wet bias of ERA5_CNN in the manuscript. It is expected that precipitation from interpolation of gauge observations or satellite-gauge merged products have small biases in low altitudes but large biases in high altitudes because rain gauges are usually located at low altitudes and have poor spatial representativeness. In our study, the ERA5_CNN is used, which is an atmospheric model-based dataset and is not corrected with rain gauge data. Constrained by the physical consistency of the atmospheric model, it is expected that ERA5_CNN generally shows consistent overestimation or underestimation at different altitudes in a basin. Aa shown in Figure R2, ERA5_CNN overestimates precipitation at most rain gauges over the TP, but

ERA5_CNN is skillful in representing the spatial variability of precipitation. As shown in Figure R3, ERA5_CNN presents more fine spatial structure of precipitation on the edge of the TP compared with IMERG. Moreover, the atmospheric model-based ERA5_CNN and HAR V2 can better represent precipitation distribution in the Karakorum of the western TP where high amount of solid precipitation is dominated, which was demonstrated in the work of Li et al. (2020, Characterizing precipitation in high altitudes of the western Tibetan plateau with a focus on major glacier areas. Int. J. Climatol. 1–14.). In general, although ERA5_CNN is biased, so far it is perhaps the best choice to characterize the precipitation gradients over the TP. In the revised manuscript, we will introduce more about the evaluation and biases in ERA5_CNN.

[Figure]

**Figure R3** Spatial pattern of annual average precipitation from (a) ERA5_CNN, (b) HAR V2 and (c) IMERG during 2008-2018. The red ovals represent the western TP where solid precipitation is dominated. This Figure will be added to the revised manuscript.

---

## Author Comment (AC3)

This study throws up an interesting topic on the precipitation gradients in the Third Pole. However, the presentation of the manuscript is rather rough. Some conclusions drawn from RPGs seem to be unreasonable. The manuscript needs to be further improved. My comments are shown as follows:

**General response:** Many thanks to the reviewer for the comments and we considered all the comments carefully. We believe these comments are helpful to improve our work. Responses to all comments are detailed below and further revisions will be given in the revised manuscript. We hope these responses have addressed your major concerns.

1. The conclusions in Figure 2 are subjective. It is difficult to conclude that ERA5_CNN is better than the other two products. In Figure 2(b) and (e), the conclusion that ERA5_CNN is the most consistent with rain gauge data is clear. However, in the other sub-basins, the conclusion is not obvious. An indicator to describe the goodness of ERA5_CNN may help.

**Response:** Thanks for the suggestion. To quantify the performance of these datasets to reflect observed spatial variability of precipitation, the spatial correlation coefficients for these datasets against gauge observations were calculated. As shown in Figure R1, in most sub-regions, the ERA5_CNN has the highest spatial correlation coefficients with gauge observations, therefore, we can conclude that ERA5_CNN can generally better reflect the observed spatial variability of precipitation than the other two products. These results will be added to the revised manuscript.

[Figure]

**Figure R1** Comparison between the altitude dependence of relative precipitation from ERA5_CNN, IMERG and HAR V2 and that from gauge observations in five networks. $P/P'$ denotes the ratio of precipitation amount ($P$) in each elevation zone to the mean precipitation amount ($P'$) at all gauge locations. The numbers within these figures represent the correlations between precipitation from gauge observations and the three datasets. "*" represents correlations significant at the 95% confidence level.

2. Are the sub-basins used in this study reasonable? It has been mentioned in the manuscript that the

precipitation decreases with altitude above 2500 m. In a sub-basin, the altitude can change from below 2500m to above 2500m. As a result, the precipitation gradients in a sub-basin are not consistent. It may need more discussion on the basin-scale precipitation gradients.

**Response:** This comment is very thought-provoking. The precipitation gradient is fitted using all grids with a specific sub-basin, therefore, the precipitation gradient is scale-dependent. Accordingly, we investigated and compared precipitation gradients at different spatial scales, as shown in Figures R2 and R3. It can be seen that precipitation gradients calculated at different spatial scales differ greatly. In addition, we can find that precipitation gradients tend to be larger and more likely to be positive at smaller spatial scales. These results will be discussed in detail in the revised manuscript.

[Figure]

**Figure R2** Spatial patterns of RPGs calculated at different sub-basin levels. The spatial scales of sub-basins (i.e. sub-basin area) generally decrease from L4 to L8.

[Figure]

**Figure R3** Comparison of (a) RPGs and (b) correlations between precipitation and altitude within each sub-basin at different sub-basin levels. Each box represents the distribution of RPGs or correlations of all the sub-basins over the TP.

3. As the numbers of gird cells in different sub-basins are different, the same values of R in different sub-basins have a different mean. For instance, R with the value of 0.5 may mean a weak correlation in a 10-grid-cell sub-basin but a strong correlation in a 100-grid-cell sub-basin. Significance tests are necessary to show the strong correlations between precipitation and altitudes.

Response: Thanks for the suggestion. We tested the significance of the correlation (Figure R4). It can be found that most sub-basins passed the 95% significance tests for the correlations. This figure will be added to the revised manuscript.

[Figure]

**Figure R4** Spatial pattern of correlations between the annual average precipitation from 1980 to 2018 and altitude for all grids within each basin. The dots represent correlations significant at the 95% confidence level.

4. In Section 4.3.1, more evidence is needed to support that strong seasonal variation exists in RPGs. The RPG is a value that the absolute precipitation gradient divided by the basin mean precipitation. The RPG will show a strong seasonal variation even if the absolute precipitation gradient has not changed. The strong seasonal variation in RPG exists but may not have any meaning.

**Response:** Affected by the monsoon climate, precipitation over the TP has a strong seasonal cycle with generally large precipitation amount in summer but small in winter. This leads to a strong seasonal cycle in the absolute precipitation gradients, as shown in Figure R5. Comparing the absolute precipitation gradients provides little information about the spatiotemporal variability of

precipitation-altitude relations because the magnitude of the absolute precipitation gradient contains the intrinsic seasonal variability of precipitation.

[Figure]

**Figure R5** Spatial patterns of the absolute precipitation gradients (PGs) in (a) winter (December to February), (b) spring (March to May), (c) summer (June to August) and (d) autumn (September to November). The PGs are calculated based on seasonal precipitation averaged from 1980 to 2018.

5. Why do the authors use the average RPGs of the five sub-regions to study the interannual variations? The interannual variations of RPGs in some sub-basins may be covered. It does not make sense to average RPGs of the sub-basins to represent the RPG of a sub-region.
**Response:** Thanks for the comment. Accordingly, we calculated the Coefficient of Variation (CV) and trends for RPGs at each sub-basin and shown in Figure R6, which allows us to analyze the interannual variations of RPGs in each sub-basin. This figure will be added to the revised manuscript and discussed.

[Figure]

**Figure R6** Spatial distribution of (a) the Coefficients of Variation (CVs) and (b) trends for annual RPGs during the period from 1980 to 2018. CVs and trends at sub-basins with missing RPGs during the analysis period were filled with white color. The dots represent trends significant at the 95% confidence level.

6. Where are the CV of annual RPGs for the sub-regions? The results should be shown in the manuscript. As RPG is a percentage, it is necessary to clarify the unit of CV. With the value of CV less than 0.12, it does not account for the conclusion that RPGs change little between different years. For example, the maximum and minimum values of RPGs in Qaidam are ~9% and ~13% respectively. Considering the range of RPGs, the change is not little. Moreover, it can be seen that there is a periodic variation in RPGs in Figure 5.

**Response:** Sorry that the Coefficient of Variation (CV) was not defined in the manuscript. The CV is dimensionless and can be calculated as follows:

$$CV = \frac{\sigma}{\mu}$$

where $\sigma$ and $\mu$ are the standard deviation and mean of a series of RPGs, respectively. The closer the CV value is to zero, the smaller the dispersion is.

A CV value of 0.12 represents that the standard deviation of RGPs during the analysis period is equal to 12% of the mean value of these RPGs. The definition of CV will be detailed in the revised manuscript.

7. The trend tests are not found in the manuscript. How to draw a conclusion that there is no significant trend in RPGs in all the sub-regions?

**Response:** Thanks for the comment. We conducted trend test for RPGs at each sub-basin and shown in Figure R6. It can be found that the trends for annual RPGs at most sub-basins are between ±0.04% and do not pass the 95% significance tests. Therefore, that there is no significant trend in RPGs in most sub-basins is accepted. Figure R6 and the corresponding analysis will be added to the revised manuscript.

8. Because of the equation RPG=a/P and the positive correlation between P and RH, there is an

inverse proportional relationship, rather than a linear relationship between RPG and RH. This analysis in Figure (a)-(e) does not make sense.

**Response:** We investigated the relations between precipitation from ERA5_CNN and relative humidity from ERA5 and shown in Figure R7. It can be found that the positive correlations between P and RH are not obvious. However, Figure 6 in the manuscript shows that there are good correlations between RPG and RH, which indicates that RH is indeed a determinant of RPG.

In addition, the motivation for analyzing the relations between RPG and RH was not clarified clearly in the manuscript. By analyzing the relations between RPGs and meteorological factors, we hope that the RPGs can be estimated empirically according to the meteorological conditions, which can broaden the implication of this study. From this perspective, any factor that has good correlations with RPG can be used. We will further clarify the motivation for analyzing the relations between RPG and RH in the revised manuscript.

[Figure]

**Figure R7** Relations between basin-average precipitation (P) and relative humidity (RH) in different periods. P and RH were averaged from 1980 to 2018.

---

## Author Response (AR1)

**Response to the Reviewers**

**General response**: Thank you for the reviewers' comments and these comments are critical for improving this work. All comments are considered and addressed carefully. The comments are given in black typeface and the authors' responses are given in blue typeface. Line numbers refer to those in the revised manuscript. Changes in the manuscript are shown in **Blue color.**

**Response to the comments from Reviewer 1# Round 1**

This study applies three datasets, ERA5_CNN, IMERG, and HAR to study precipitation gradients in the Third Pole region. Instead of the absolute precipitation gradient, this study uses the relative precipitation gradient (RPG) throughout the manuscript. However, I think the RPG is not an appropriate index which can cause misleading and even wrong results. This problem makes the study questionable. My comments are as below.

1. RPG problems

The definition of relative precipitation gradient (RPG) in Equation (2) is problematic from my perspective. There is no obvious linkage between precipitation intensity and precipitation gradient because precipitation gradient is just a spatial pattern no matter how large the mean value is (e.g.,1 0 or 1000). The authors state that RPG is used because ERA5_CNN has systematic bias in the TP. The most common systematic bias is overestimation or underestimation. If we assume the spatial distribution of overestimation or underestimation is even (cases with uneven distributions are more complex), the precipitation gradient has nothing to with the magnitude of precipitation, which means the normalization operation in Equation (2) is meaningless. Actually, the normalization could be harmful. For example, Let's assume a is 0.2 mm/m and P is 2 mm/day according to ERA5_CNN. Then, RPG=a/P=0.2/2 = 0.1. If we have another dataset which could have similar spatial distribution with ERA5_CNN but an overestimation bias. Let's say its P=4 mm/day. According to results from this study, the gradient for this new dataset will be 0.1 X 4 =0.4, which is incorrect. I can imagine that this simple example could be quite common in applications motivated by this work.

The RPG problem could also result in wrong comparison between different products. For example, when comparing ERA5_CNN and gauge data in a basin, RPG cannot provide useful information. Even the RPG values of the two products are the same, their absolute precipitation gradients are often different because their mean precipitation is different. Similarly, even the RPG of a product is biased, its absolute precipitation gradient could be correct. Therefore, the use of RPG does not make sense.

Besides, I recommend the authors also present absolute precipitation gradient. RPG maps cannot show the impact of atmosphere vapor. For example, large RPG may happen in dry basins, but its significance

could be weaker compared to relatively smaller RPG in wet basins where precipitation amounts and bias are larger.

**Response:** Response: We thank the reviewer for the comments, which are helpful for us to improve the quality of this manuscript.

1) The main concern of the reviewer is whether the RPG (relative precipitation gradient) is rational. Maybe our presentation had given you the impression that RPG is more rational than APG (absolute precipitation gradient). This is not always the case, and thank you for pointing out this. In the revised manuscript, we have presented both APG and RPG that may be favorable for the readers. With your suggestion, we conducted further analyses, and here we try to clarify that RPG is less sensitive to model biases and to climate change:

One of the main objectives of this study is to obtain spatially distributed precipitation gradients for assisting the interpolation of rain gauge data in the TP. In this case, the interpolation may be conducted with either APG or RPG. So, the question is: which one of APG or RPG is more applicable? If the absolute biases in ERA5_CNN are more uniform in space than the relative biases, the APG derived from ERA5_CNN is more accurate. If the relative biases in ERA5_CNN are more uniform in space than the absolute biases, the RPG is more accurate and applicable. According to the evaluation results shown in Figure R1.1 (i.e. Figure 7 in the revised manuscript), the spatial CV (coefficient of variation) values for Abias and Rbias are 1.59 and 1.24, respectively. The relative biases in ERA5_CNN are indeed more homogeneous than the absolute biases. Therefore, the RPG is expected to be less sensitive to bias in precipitation products and more applicable for assisting interpolation of rain gauge data.

[Figure]

**Figure R1.1** Spatial patterns of (a) absolute bias (Abias) and (b) relative bias (Rbias) for annual precipitation from ERA5_CNN during 1980-2018 at CMA stations.

In addition, it can be seen from Figure R1.2 (i.e. Figure 6 in the revised manuscript) that APG shows significant positive trends at many sub-basins of the TP, mainly because the TP generally become wetter

in recent decades. However, the trend in RPG is not significant. This suggests that APG is more sensitive to precipitation amount under climate change than RPG, and thus the RPG obtained in a certain period is expected to be more representative than APG when applying for precipitation interpolation under climate change.

[Figure]

**Figure R1.2** Spatial distribution of (a) and (b) the coefficient of variation (CV) and (c) and (d) trend for annual APGs and RPGs during 1980 to 2018.

2) The second concern is the comparison of RPG between different products or in different periods. The APG is related to the precipitation amount. For example, the APGs calculated based on HAR V2 are generally larger than those from the ERA5_CNN (Figure R1.3), because HAR V2 has a larger precipitation amount in the TP than the ERA5_CNN. In addition, the APG in summer is remarkably large than that in other seasons, due to the seasonal cycle of precipitation amount. Therefore, the APG may be an indicator of precipitation amount and comparing APG from different datasets or in different seasons provides little information.

In the comment, the reviewer gave an example that the same RPG value will lead to different absolute gradients when datasets with different mean precipitation are used. This is true, but the differences in absolute precipitation gradients were caused by the differences in precipitation mean, rather than RPG. In reality, we expect that RPG will be used with the mean from observations when applying for interpolating rain gauge data.

[Figure]

**Figure R1.3** Spatial pattern of annual absolute precipitation gradients (APGs; left panel) and relative precipitation gradients (RPGs; right panel) from ERA5_CNN and HAR V2. This figure is not essential in this study, thus, it is not presented in the revised manuscript.

85

Accordingly, several revisions were made to the revised manuscript. 1) We further clarified the objectives of this study, please refer to Line 50-53 and 86-88 in the revised manuscript. 2) we added more evaluations and discussions of the biases in ERA5_CNN and the uncertainties in both APG and RPG, please refer to Figure R1.1 (Figure 7 in the revised manuscript) and section 5.1 in the revised

90 manuscript. 3) Given that the characteristics of biases in different datasets may vary greatly, comparing the altitude dependence of relative precipitation from different datasets is complex, we compared the altitude dependence of absolute rather than relative precipitation amount from different datasets in the revised manuscript, please refer to Figure 2 (i.e. Figure R1.4 in the response) and section 4.1 in the revised manuscript. 4) We have presented both APG and RPG in the revised manuscript.

95

[Figure]

**Figure R1.4** Comparison between the altitude dependence of precipitation from ERA5_CNN, IMERG and HAR V2 and that from rain gauge data in five networks. The lines show the average precipitation amount in each altitude zone and the bars denote the number of rain gauges in each zone.

2. Atmospheric model still has large bias in complex terrain including the study area which is also acknowledged in this study. This problem needs more discussion.

**Response**: Thanks for the comment! We have included more evaluations and discussions about the uncertainties in the atmospheric model-based ERA5_CNN dataset in the revised manuscript. Please refer to Figure 7 (i.e. Figure R1.1 in the response) and section 5.1 in the revised manuscript.

3. The discussion between the relationship between precipitation and wind speed can be further improved. They are both aspects of air mass movement which is affected by both atmosphere and topography. If you try to investigate the causality (e.g., some sentences in abstract and discussion parts), you should go further. High or low wind speed is also the result of various environmental factors.

**Response**: We absolutely agree that topography has impact on the spatial variability of both precipitation and wind speed. In the revised manuscript, we discussed the relations between precipitation gradients

and wind speed/relative humidity within different sub-regions of the TP that may have diverse
topography. As shown in Figure R1.5 (i.e. Figure 8 in the revised manuscript), in different sub-regions,
the relations between RPG and wind speed (relative humidity) are similar, therefore, the conclusion that
RPG has positive (negative) correlations with wind speed (relative humidity) is robust. Corresponding
analysis was added to the revised manuscript (Line 351-357 and Line 363-367). Because the mechanism
for the variations of precipitation gradients is complex, whose exploration is beyond the scope of this
study, we only discuss the possible factors that may be related to the variations rather than give the
causality.

[Figure]

**Figure R1.5** Relationships between annual RPGs and (a-f) basin-average relative humidity (RH) and (g-
l) wind speed (Va) in different sub-regions of the TP. ETP: eastern TP; YTR: Yarlung Tsangpo River
Basin; ITP: Inner TP; QDM: Qaidam Basin; UID: Upper Indus

4. Section 2.1: what's the time period of the 1/30-degree WRF simulations? Since ERA5_CNN is taken
from Jiang et al. (2021), it is recommended to introduce more method and evaluation details in the
manuscript.

**Response**: The 1/30-degree WRF simulations cover the whole years of 2013 and 2018. We have added
more details about the production and previous evaluation of ERA5_CNN in the revised manuscript.
Please refer to Line 102-107 and 112-114 in the revised manuscript.

5. Section 2.2 introduces IMERG and HAR V2 which should be in "Section 2.1 Precipitation Datasets".

**Response**: Thanks for the suggestion! The introductions of IMERG and HAR V2 were moved to Section
2.1.

6. Equation (1): units of P and H are needed.

**Response**: The units of P and H are "mm.day$^{-1}$" and "100 m", respectively. We have added them in the revised manuscript. Please see Line 148 in the revised manuscript.

7. Line 145-149: This description has a logical problem. It is stated that "the precipitation gradient is estimated only when the following three principles are met" and the third principle is "(3) the p-value of the Student's t-test for the regression equation should be less than 0.05". But you cannot know p-value without estimating precipitation gradient. Please revise this sentence.

**Response**: Thanks for pointing out this unclear description! Actually, we first fit the regression equation at each sub-basin, then, the Student's t-test for the regression equation is conducted. If the p-value of the Student's t-test is less than 0.05, the slope of the fitted line is treated as the precipitation gradient; otherwise, the precipitation of the sub-basin is filled with a missing value. We revised this sentence as "The value of precipitation gradient for a sub-basin is given only when the following three principles are met: 1) the number of grids within the sub-basin should not be less than 10; 2) the standard deviation of altitude within the sub-basin should not be less than 50 m; 3) the p-value of the Student's t-test for the regression equation should be less than 0.05." Please refer to Line 157-160 in the revised manuscript.

8. Figure 2: I suggest adding rain gauge number of different elevation bands.

**Response**: Thanks for the suggestion, the number of rain gauges in each elevation band was added in Figure 2 (i.e. Figure R1.4 in the response) in the revised manuscript.

9. Is Tibetan Plateau a more suitable word compared to "Third Pole"? Tibetan Plateau is clearer as a geographical concept.

**Response**: Our study region covers a wide region, including Pamir and Hindukush. This region is now called Third Pole.

10. The writing and presentation of the manuscript can be further improved.

**Response**: Thanks for the suggestion! We have thoroughly revised and edited the language of the manuscript.

**Response to the comments from Reviewer 1 #Round 2**

Thank you for the quick response. The responses to all comments look great to me, except for the RPG problem. I don't think the explanation of systematic bias can solve the problem. Before I state my reasons, I will just give my suggestion: the authors can include both relative and absolute precipitation gradients and thoroughly discuss absolute VS relative gradients in the manuscript. The current manuscript only has

five figures presenting the quantitative results (Figures 2 to 6). I believe as a research article, it has enough room to include more results which will make this paper more interesting and informative. The comparison between absolute and relative gradients can partly solve the concerns, considering gradients from ERA5_CNN contain large uncertainties in the third pole.

The authors' explanation is that the systematic bias can be expressed as the relative bias (i.e., a fraction of precipitation amount), which is relatively uniform in different regions of the TP. Therefore, it is appropriate to calculate RPG. However,

1. The experiments in my previous comments are not answered. RPG from different datasets/regions/periods is not comparable. For example, for cases where RPG1 from ERA5_CNN and RPG2 from rain gauge data are the same, we cannot say RPG1 is perfect or not because ERA5_CNN and RPG2 could have different mean precipitation. On the other hand, if RPG1 and RPG2 are different, it is still possible that ERA5_CNN captures the correct gradient pattern. Besides, the signs of RPG under/over estimation could be different from under/over estimation of absolute precipitation gradients, making the results-based RPG less reliable. Due to this problem, evaluation of ERA5_CNN using rain gauge data and comparing gradients in different regions of the third pole in the manuscript could be meaningless using RPG.

**Response:** According to the reviewer's suggestion, we have presented both APG and RPG in the revised manuscript. The reasons for presenting the RPG were given in the above response (we hope we have understood your concern), please refer to the response to RPG problems above. Moreover, when comparing ERA5_CNN, HAR V2, IMERG and rain gauge data, as suggested, the altitude dependence of absolute rather than relative precipitation amount was presented in the revised manuscript, as shown in Figure 2 (i.e. Figure R1.4 in the response) and section 4.1 in the revised manuscript.

2. There is no evidence that the relative bias is uniformly distributed in space. Relative bias is affected by many factors particularly in the large scale, while precipitation amount is just one of those factors. Actually, if relative bias can be so easily estimated, bias correction should be an easy task such as in the third pole, but the reality is that researchers are struggling with bias correction in complex terrain. I believe the authors hope that the RPG calculated in this study can be applied in other situations, but if the RPG is built on assumptions with large uncertainties, the application of RPG will be risky.

**Response:** We agree that the relative biases in ERA5_CNN vary in space and that biases in ERA5_CNN will result in uncertainties in RPG. There is a long way to go to achieve a high-accuracy estimation for either RPG or APG, and what we can do is to gradually improve our understanding. What we can say is that it is relatively uniform than the absolute bias, as shown in Figure R1.1 (i.e. Figure 7 in the revised manuscript), which makes the RPG more applicable for assisting precipitation interpolation, as clarified

210

in the above response. In the revised manuscript, we have given more evaluations about the biases in the ERA5_CNN and more discussions on the uncertainties in both APG and RPG. Please refer to section 5.1 in the revised manuscript.

3. The definition of bias is unclear. In evaluation studies, the relative bias is calculated against the reference dataset such as ground observations, but the calculation of RPG in this study is against the target dataset ERA5_CNN. I don't know how large the impact is, but this can weaken the reliability of RPG. For example, for a mountain slope, ERA5_CNN has low precipitation (P1) in low elevation and

215 high precipitation (P2) in high elevation, I expect that P1 is more reliable than P2 because models are less reliable in high elevation. Using the method in this study, we can calculate RPG1 in low elevation and RPG2 in high elevation. Comparing the quality of RPG1 and RPG2 is cumbersome because we don't the direction (over or underestimation) of P1 and P2. Of course, this problem also affects absolute gradients, but after normalizing using P1 and P2, this problem becomes too complex.

220 **Response:** We have given more details about the bias of ERA5_CNN in the revised manuscript (section 5.1). It is expected that precipitation from interpolation of gauge observations or satellite-gauge merged products have small biases in low altitudes but large biases in high altitudes because rain gauges are usually located at low altitudes and have poor spatial representativeness. In our study, the ERA5_CNN is used, which is an atmospheric model-based dataset and is not corrected with rain gauge data.

225 Constrained by the physical consistency of the atmospheric model, ERA5_CNN is expected to be overall consistent overestimation or underestimation at different altitudes in a basin. As shown in Figure R1.1 (i.e. Figure 7 in the revised manuscript).

**Response to the comments from Reviewer 2**

This study throws up an interesting topic on the precipitation gradients in the Third Pole. However, the presentation of the manuscript is rather rough. Some conclusions drawn from RPGs seem to be unreasonable. The manuscript needs to be further improved. My comments are shown as follows:

1. The conclusions in Figure 2 are subjective. It is difficult to conclude that ERA5_CNN is better than the other two products. In Figure 2(b) and (e), the conclusion that ERA5_CNN is the most consistent with rain gauge data is clear. However, in the other sub-basins, the conclusion is not obvious. An indicator to describe the goodness of ERA5_CNN may help.

**Response:** Thanks for the suggestion. To quantify the performance of these datasets to reflect observed spatial variability of precipitation, the spatial correlation coefficients for these datasets against gauge observations were calculated. As shown in Figure R2.1 (i.e. Figure 2 in the revised manuscript), in most sub-regions, the ERA5_CNN has the highest spatial correlation coefficients with gauge observations, therefore, we can conclude that ERA5_CNN can generally better reflect the observed spatial variability of precipitation than the other two products. These results were added to the revised manuscript (section 4.1).

[Figure]

**Figure R2.1** Comparison between the altitude dependence of precipitation from ERA5_CNN, IMERG and HAR V2 and that from rain gauge data in five networks. The lines show the average

2. Are the sub-basins used in this study reasonable? It has been mentioned in the manuscript that the precipitation decreases with altitude above 2500 m. In a sub-basin, the altitude can change from below 2500m to above 2500m. As a result, the precipitation gradients in a sub-basin are not consistent. It may need more discussion on the basin-scale precipitation gradients.

**Response:** This comment is very thought-provoking. The precipitation gradient is fitted using all grids with a specific sub-basin, therefore, the precipitation gradient is scale-dependent. Accordingly, we investigated and compared precipitation gradients at different spatial scales, as shown in Figures R2.2 and R2.3 (i.e. Figures 9 and 10 in the revised manuscript). It can be seen that precipitation gradients calculated at different spatial scales differ greatly. In addition, we can find that precipitation gradients tend to be positive at smaller spatial scales. As the spatial scale decreases, the precipitation gradient first increases and then remains relatively stable. These results were discussed in section 5.3 in the revised manuscript.

[Figure]

**Figure R2.2** Spatial patterns of annual RPGs calculated at different sub-basin levels. The spatial scales of sub-basins (i.e. sub-basin area) generally decrease from L4 to L7.

[Figure]

**Figure R2.3** Comparison of (a) RPGs and (b) correlations between precipitation and altitude calculated at different sub-basin levels. Each box represents the distribution of RPGs or correlations of all the sub-basins over the TP.

3. As the numbers of gird cells in different sub-basins are different, the same values of R in different sub-basins have a different mean. For instance, R with the value of 0.5 may mean a weak correlation in a 10-grid-cell sub-basin but a strong correlation in a 100-grid-cell sub-basin. Significance tests are necessary to show the strong correlations between precipitation and altitudes.

Response: Thanks for the suggestion. We tested the significance of the correlation and shown in Figure R2.4 (Figure 3a in the revised manuscript). It can be found that most sub-basins passed the 95% significance tests for the correlations.

[Figure]

**Figure R2.4** Spatial distribution of the correlations between the annual average precipitation and altitude for all grids within each basin.

4. In Section 4.3.1, more evidence is needed to support that strong seasonal variation exists in RPGs. The RPG is a value that the absolute precipitation gradient divided by the basin mean precipitation. The RPG will show a strong seasonal variation even if the absolute precipitation gradient has not changed. The strong seasonal variation in RPG exists but may not have any meaning.

**Response:** In the revised manuscript, we presented both absolute precipitation gradient (APG) and relative precipitation gradient (RPG). Affected by the monsoon climate, precipitation over the TP

has a strong seasonal cycle with generally large precipitation amount in summer but small in winter. This leads to a strong seasonal cycle in the absolute precipitation gradients, as shown in Figure R2.5 (i.e. Figure 4 in the revised manuscript).

As shown in Figure R2.5 and Figure R2.6, most of the sub-basins in the Himalayas have positive precipitation gradients in winter, however, this region is dominated by negative gradients in summer. In spring and autumn, the western Himalayas has positive gradients and the eastern Himalayas has negative gradients. These results indeed imply that precipitation gradients have remarkable seasonal variations and were introduced in Line 281-285 in the revised manuscript. Besides, many previous works also show the seasonal variations of precipitation gradients, which are discussed in Line 285-289 in the revised manuscript.

[Figure]

**Figure R2.5** Spatial distribution of APGs in (a) winter (December to February), (b) spring (March to May), (c) summer (June to August) and (d) autumn (September to November). The RPGs are calculated based on seasonal precipitation averaged from 1980 to 2018.

[Figure]

**Figure R2.6** Same as Figure R2.5 but for RPGs.

5. Why do the authors use the average RPGs of the five sub-regions to study the interannual variations? The interannual variations of RPGs in some sub-basins may be covered. It does not make sense to average RPGs of the sub-basins to represent the RPG of a sub-region.

**Response:** Thanks for the comment! Accordingly, we calculated the Coefficient of Variation (CV) and trends for RPG at each sub-basin, as shown in Figure R2.7 (i.e. Figure 6 in the revised manuscript), which allows us to analyze the interannual variations of RPGs in each sub-basin. Corresponding analysis was added to section 4.3.2 in the revised manuscript.

[Figure]

**Figure R2.7** Spatial distribution of (a) and (b) the coefficient of variation (CV) and (c) and (d) trend for

annual APGs and RPGs during 1980 to 2018.

6. Where are the CV of annual RPGs for the sub-regions? The results should be shown in the manuscript. As RPG is a percentage, it is necessary to clarify the unit of CV. With the value of CV less than 0.12, it does not account for the conclusion that RPGs change little between different years. For example, the maximum and minimum values of RPGs in Qaidam are ~9% and ~13% respectively. Considering the range of RPGs, the change is not little. Moreover, it can be seen that there is a periodic variation in RPGs in Figure 5.

**Response:** Sorry that the Coefficient of Variation (CV) was not defined in the manuscript. The CV is dimensionless and can be calculated as follows:

$$CV = \frac{\sigma}{|\mu|}$$

where $\sigma$ and $|\mu|$ are the standard deviation and absolute mean of a series of samples, respectively. The closer the CV value is to zero, the smaller the dispersion is. The definition of CV was added to the revised manuscript (Line 173-177 in the revised manuscript). In addition, the CV values for annual APG and RPG are shown in Figure 6 (i.e. Figure R2.7 in the response) and discussed in Line 297-298 of section 4.3.2 in the revised manuscript.

7. The trend tests are not found in the manuscript. How to draw a conclusion that there is no significant trend in RPGs in all the sub-regions?

**Response:** Thanks for the comment! We conducted trend tests for both APG and RPG at each sub-basin and shown in Figure R2.7 (i.e. Figure 6 in the revised manuscript). It can be found that APG has positive trends at most sub-basins, especially in the Inner TP, which is mainly because precipitation amount in the TP has generally increased in recent decades. However, the trends of annual RPG at most sub-basins do not pass the 95% significance tests. Figure R2.7 and the corresponding analysis were added to the revised manuscript. Please refer to section Figure 6 and Line 298-307 of section 4.3.2 in the revised manuscript.

8. Because of the equation RPG=a/P and the positive correlation between P and RH, there is an inverse proportional relationship, rather than a linear relationship between RPG and RH. This analysis in Figure (a)-(e) does not make sense.

**Response:** We investigated the relations between precipitation from ERA5_CNN and relative humidity from ERA5 and shown in Figure R2.8. It can be found that the positive correlations between P and RH are not obvious. However, our results in the manuscript show that there are good

correlations between RPG and RH, which indicates that RH is indeed a determinant of RPG.

We are sorry that the motivation for analyzing the relations between RPG and RH was not clarified clearly in the manuscript. By analyzing the relations between RPGs and meteorological factors, we hope that the RPGs can be estimated empirically according to the meteorological conditions, which can broaden the implication of this study. From this perspective, any factor that has good correlations with RPG can be used. We have introduced the motivation for analyzing the relations between RPG and RH in Line 92-95 in the revised manuscript.

[Figure]

**Figure R2.8** (a) Relations between annual RPG and basin-average relative humidity (RH). (b) Relations between basin-average annual precipitation (P) and relative humidity (RH). P and RH were averaged from 1980 to 2018. RPG is calculated based on annual average precipitation from 1980 to 2018. Because this figure is not in the scope of this study, it is not shown in the manuscript.

---

## Editor Decision (ED1)

Hydrol. Earth Syst. Sci. Discuss., referee comment RC1
https://doi.org/10.5194/hess-2022-103-RC1, 2022

[Figure]

**Comment on hess-2022-103**

Anonymous Referee #1

Referee comment on "Characterizing basin-scale precipitation gradients in the Third Pole region and associated determinants" by Yaozhi Jiang et al., Hydrol. Earth Syst. Sci. Discuss., https://doi.org/10.5194/hess-2022-103-RC1, 2022

This study applies three datasets, ERA5_CNN, IMERG, and HAR to study precipitation gradients in the Third Pole region. Instead of the absolute precipitation gradient, this study uses the relative precipitation gradient (RPG) throughout the manuscript. However, I think the RPG is not an appropriate index which can cause misleading and even wrong results. This problem makes the study questionable. My comments are as below.

1. RPG problems

The definition of relative precipitation gradient (RPG) in Equation (2) is problematic from my perspective. There is no obvious linkage between precipitation intensity and precipitation gradient because precipitation gradient is just a spatial pattern no matter how large the mean value is (e.g.,1 0 or 1000). The authors state that RPG is used because ERA5_CNN has systematic bias in the TP. The most common systematic bias is overestimation or underestimation. If we assume the spatial distribution of overestimation or underestimation is even (cases with uneven distributions are more complex), the precipitation gradient has nothing to with the magnitude of precipitation, which means the normalization operation in Equation (2) is meaningless. Actually, the normalization could be harmful. For example, Let's assume a is 0.2 mm/m and P is 2 mm/day according to ERA5_CNN. Then, RPG=a/P=0.2/2 = 0.1. If we have another dataset which could have similar spatial distribution with ERA5_CNN but an overestimation bias. Let's say its P=4 mm/day. According to results from this study, the gradient for this new dataset will be 0.1 X 4 =0.4, which is incorrect. I can imagine that this simple example could be quite common in applications motivated by this work.

The RPG problem could also result in wrong comparison between different products. For example, when comparing ERA5_CNN and gauge data in a basin, RPG cannot provide useful information. Even the RPG values of the two products are the same, their absolute

precipitation gradients are often different because their mean precipitation is different. Similarly, even the RPG of a product is biased, its absolute precipitation gradient could be correct. Therefore, the use of RPG does not make sense.

Besides, I recommend the authors also present absolute precipitation gradient. RPG maps cannot show the impact of atmosphere vapor. For example, large RPG may happen in dry basins, but its significance could be weaker compared to relatively smaller RPG in wet basins where precipitation amounts and bias are larger.

2. Atmospheric model still has large bias in complex terrain including the study area which is also acknowledged in this study. This problem needs more discussion.

3. The discussion between the relationship between precipitation and wind speed can be further improved. They are both aspects of air mass movement which is affected by both atmosphere and topography. If you try to investigate the causality (e.g., some sentences in abstract and discussion parts), you should go further. High or low wind speed is also the result of various environmental factors.

4. Section 2.1: what's the time period of the 1/30-degree WRF simulations? Since ERA5_CNN is taken from Jiang et al. (2021), it is recommended to introduce more method and evaluation details in the manuscript.

5. Section 2.2 introduces IMERG and HAR V2 which should be in "Section 2.1 Precipitation Datasets".

6. Equation (1): units of P and H are needed.

7. Line 145-149: This description has a logical problem. It is stated that "the precipitation gradient is estimated only when the following three principles are met" and the third principle is "(3) the p-value of the Student's t-test for the regression equation should be less than 0.05". But you cannot know p-value without estimating precipitation gradient. Please revise this sentence.

8. Figure 2: I suggest adding rain gauge number of different elevation bands.

9. Is Tibetan Plateau a more suitable word compared to "Third Pole"? Tibetan Plateau is clearer as a geographical concept.

10. The writing and presentation of the manuscript can be further improved.

the value of CV less than 0.12, it does not account for the conclusion that RPGs change little between different years. For example, the maximum and minimum values of RPGs in Qaidam are ~9% and ~13% respectively. Considering the range of RPGs, the change is not little. Moreover, it can be seen that there is a periodic variation in RPGs in Figure 5.

- The trend tests are not found in the manuscript. How to draw a conclusion that there is no significant trend in RPGs in all the sub-regions?
- Because of the equation RPG=a/P and the positive correlation between P and RH, there is an inverse proportional relationship, rather than a linear relationship between RPG and RH. This analysis in Figure (a)-(e) does not make sense.

---

## Author Response (AR2)

**General response**: Thank you for the reviewer's comments. The comments are given in **black typeface** and the authors' responses are given in **blue typeface**. Line numbers refer to those in the revised manuscript. Changes in the manuscript are shown in **Blue color.**

The authors addressed most of my comments. Based on that, I suggest a couple of minor points to be revised.

1. Some explanation is needed in the caption of Figure 1, for example, the meaning of the numbers and the signal "*".

Response: Thanks for pointing out this. We thought the reviewer means Figure 2 rather than Figure 1. The numbers in Figure 2 are the spatial correlations of precipitation amount between rain gauge data and precipitation products. The signal "*" represents that the correlation is significant at the 95% confidence level. These contents were added to the caption of Figure 2 in revised manuscript (Line 188-190).

2. Line 320-325. "According to the definition of APG and RPG, if Abias is spatially homogeneous, the APG in this study is equal to that derived from rain gauge data, and if Rbias is uniform in space, the RPG in this study is consistent with that from rain gauge data." This sentence is difficult to understand for me. Therefore, the results from Figure 7 need to be further interpreted.

**Response:** Sorry that this sentence is not very clear. We have added a schematic and relevant formulas to further clarify this. As shown in Figure R1, if absolute bias (Abias) is spatially homogeneous (i.e. the precipitation product has the same absolute value of overestimation or underestimation at all locations in a region), the slopes of the regression line (i.e. the APG) derived from the product (the blue line) and rain gauge data (the black line) are the same because these two lines are parallel (as shown in Figure R1); if the relative bias (Rbias) is uniform in space (i.e. the precipitation product has the same percentage of overestimation or underestimation), the calculated RPG is consistent with that from rain gauge data because the basin-average precipitation and the APG in Equation 2 have the same percentage of bias (as shown in Figure R1b). These contents were added to the revised manuscript (see Line 328-335).

[Figure]

**Figure R1** Schematic of the impact of bias in precipitation product on the calculation of APG and RPG. (a) Precipitation product with spatially homogeneous Abias. (b) Precipitation product with

spatially homogeneous Rbias. $O$ and $P$ represent precipitation from rain gauge data and biased product, respectively. $E$ is the elevation. $\bar{O}$ represents the basin-average precipitation amount of rain gauge data. The black and blue lines are the fitted regress lines between elevation and precipitation from rain gauge data and precipitation product, respectively.